

# Reconstruction of global surface ocean $p$CO$_2$ using region-specific predicators based on a stepwise FFNN regression algorithm

Guorong Zhong[1,2,3,4], Xuegang Li[1,2,3,4*], Jinming Song[1,2,3,4*], Baoxiao Qu[1,3,4], Fan Wang[1,2,3,4], Yanjun Wang[1,4], Bin Zhang[1,4], Xiaoxia Sun[1,2,3,4], Wuchang Zhang[1,3,4], Zhenyan Wang[1,3,4], Jun Ma[1,3,4], Huamao Yuan[1,2,3,4], Liqin Duan[1,2,3,4]

[1]Institute of Oceanology, Chinese Academy of Sciences, Qingdao 266071, China
[2]University of Chinese Academy of Sciences, Beijing 101407, China
[3]Pilot National Laboratory for Marine Science and Technology, Qingdao 266237, China
[4]Center for Ocean Mega-Science, Chinese Academy of Sciences, Qingdao 266071, China

*Correspondence to*: Xuegang Li (lixuegang@qdio.ac.cn); Jinming Song (jmsong@qdio.ac.cn)

**Abstract**: Various machine learning methods were attempted in the global mapping of surface ocean partial pressure of CO$_2$ ($p$CO$_2$) to reduce the uncertainty of global ocean CO$_2$ sink estimate due to undersampling of $p$CO$_2$. In previous researches the predicators of $p$CO$_2$ were usually selected empirically based on theoretic drivers of surface ocean $p$CO$_2$ and same combination of predictors were applied in all areas unless lack of coverage. However, the differences between the drivers of surface ocean $p$CO$_2$ in different regions were not considered. In this work, we combined the stepwise regression algorithm and a Feed Forward Neural Network (FFNN) to selected predicators of $p$CO$_2$ based on mean absolute error in each of the 11 biogeochemical provinces defined by Self-Organizing Map (SOM) method. Based on the predicators selected, a monthly global 1° × 1° surface ocean $p$CO$_2$ product from January 1992 to August 2019 was constructed. Validation of different combination of predicators based on the SOCAT dataset version 2020 and independent observations from time series stations was carried out. The prediction of $p$CO$_2$ based on region-specific predicators selected by the stepwise FFNN algorithm were more precise than that based on predicators from previous researches. Appling of a FFNN size improving algorithm in each province decreased the mean absolute error (MAE) of global estimate to 11.32 µatm and the root mean square error (RMSE) to 17.99 µatm. The script file of the stepwise FFNN algorithm and $p$CO$_2$ product are distributed through the Institute of Oceanology of the Chinese Academy of Sciences Marine Science Data Center (IOCAS; http://dx.doi.org/10.12157/iocas.2021.0022, Zhong et al., 2021).



## 1 Introduction

As a net sink for atmospheric $CO_2$, global oceans have been thought to have removed about one third of anthropogenic $CO_2$ since the beginning of the industrial revolution (Sabine et al., 2004; Friedlingstein et al., 2019). However, great differences existed between previous estimates of sea-air $CO_2$ flux, due to large uncertainty in estimates of surface ocean partial pressure of $CO_2$ ($p$$CO_2$) (Regnier et al., 2013; Schuster et al., 2013; Wanninkhof et al., 2013; Ishii et al., 2014). surface ocean $p$$CO_2$ is an essential parameter to describe the release and uptake for atmospheric $CO_2$ by the oceans in the data-based method. Greater $p$$CO_2$ of surface water than that of overlying air indicating that $CO_2$ released from oceans to the air, and absorption of $CO_2$ by oceans happened when the $p$$CO_2$ of surface water was lower than that of air. The ocean in these two scenarios is known as oceanic carbon source and oceanic carbon sink respectively. Sparse and uneven observations of surface ocean $p$$CO_2$ in time and space severely limited the understanding of interannual variability of oceanic carbon sink, and researches based on different methods were carried out to break this barrier. In earlier studies, traditional unitary and multiple regression methods between surface ocean $p$$CO_2$ and its drivers was attempted in the mapping of surface ocean $p$$CO_2$, which were limited in specific regions and sometimes even in specific seasons with a relatively high root mean square error (RMSE) (Sarma et al., 2006; Takahashi et al., 2006; Shadwick et al., 2010; Chen et al., 2011; Marrec et al., 2015). Recent researches on artificial neural networks and other machine learning algorithms, such as feed-forward neural network (FFNN) method (Zeng et al., 2014; Zeng et al., 2015; Moussa et al., 2016; Denvil-Sommer et al., 2019) and self-organization mapping (SOM) method (Friedrich and Oschlies, 2009; Telszewski et al., 2009; Hales et al., 2012; Nakaoka et al., 2013), significantly reduced the bias in the interpolation based on relationships between surface ocean $p$$CO_2$ and its drivers. In addition, method such as finding better predicators or combining SOM and other neural networks was also attempt to further decrease the $p$$CO_2$ predicting error (Hales et al., 2012; Nakaoka et al., 2013; Landschuetzer et al., 2014; Chen et al., 2019; Denvil-Sommer et al., 2019; Zhong et al., 2020; Wang et al., 2021). However, the selection of predicators in the surface ocean $p$$CO_2$ mapping was more empirical, focusing on the theoretical drivers of the $p$$CO_2$ and its variation. Sea surface temperature and salinity were considered as the most important and used in almost all related studies (Landschutzer et al., 2013; Nakaoka et al., 2013; Moussa et al., 2016; Laruelle et al., 2017; Zeng et al., 2017; Denvil-Sommer et al., 2019), similarly the chlorophyll-a concentration is also widely used (Nakaoka et





al., 2013; Landschuetzer et al., 2014; Laruelle et al., 2017; Zeng et al., 2017; Denvil-
Sommer et al., 2019). One more indicator, mixed layer depth, appeared frequently in
related studies (Telszewski et al., 2009; Nakaoka et al., 2013; Landschuetzer et al., 2014;
Zeng et al., 2017; Denvil-Sommer et al., 2019). Besides, the sampling information have
been also used as indicators, including latitude and longitude (Friedrich and Oschlies,
2009; Jo et al., 2012; Zeng et al., 2015; Zeng et al., 2017; Denvil-Sommer et al., 2019),
and sampling time (Friedrich and Oschlies, 2009; Zeng et al., 2015). In recent
researches, dry air mixing ratio of atmospheric $CO_2$ ($xCO_2$) was also used as a
predicator (Landschuetzer et al., 2014; Denvil-Sommer et al., 2019). The sea surface
height, which was considered effective in improving the spatial pattern and the accuracy
of surface ocean $pCO_2$ mapping at the basin and regional scale, and the monthly
anomalies of the most widely used parameters mentioned above were used by the
Denvil-Sommer et al (2019). In the research focused on the surface ocean $pCO_2$
mapping of coastal areas, the bathymetry, sea ice and wind speed were also used as
indicators (Laruelle et al., 2017). In each of these researches, same combination of
indicators was applied in all areas of the global ocean, although the global ocean was
divided into several biogeochemical provinces in some of the researches. However, the
indicator that plays an important role in the surface ocean $pCO_2$ reconstruction at one
region may be not a good predicator of surface ocean $pCO_2$ in other regions, due to
complex and variable drivers in different regions. But no widely recognized method for
judging the importance of each predicator in the surface ocean $pCO_2$ mapping are
available yet. Thus, we attempted to construct a stepwise FFNN algorithm to rank the
importance of predicators and figure out the optimal combination in each
biogeochemical province defined by SOM, for decreasing the predication errors in the
surface ocean $pCO_2$ mapping.

## 2 Methodology

### 2.1 Data

The surface ocean fugacity of $CO_2$ ($fCO_2$) observation data from the Surface Ocean
$CO_2$ Atlas $fCO_2$ dataset version 2020 (SOCATv2020) (Bakker et al., 2016) was used to
construct the non-liner relationship between surface ocean $pCO_2$ and predicators. The
transition between $fCO_2$ and $pCO_2$ was following the formula (Körtzinger, 1999):

$$f\mathrm{CO_2} = p\mathrm{CO_2} \cdot exp\left(P \cdot \frac{B+2\delta}{\mathrm{R}T}\right) \tag{1}$$

where $P$ is the total atmospheric surface pressure using the National Centers for
Environmental Prediction (NCEP) monthly mean sea level pressure product (Dee et al.,

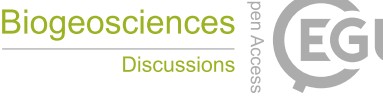

2011), and $T$ is the absolute temperature. R is the gas constant (8.314 J K$^{-1}$ mol$^{-1}$).
Parameters $B$ and $\delta$ are both viral coefficients (Weiss, 1974).
In this work, parts of indicators was choose from previous researches of surface
ocean $p$CO$_2$ reconstruction based on machine learning methods, including sea surface
temperature (SST) and sea surface salinity (SSS) using the 1°×1° gridded product from
Chen et al (2017) at http://159.226.119.60/cheng/ and the anomalies (SST$_{anom}$ and
SSS$_{anom}$), chlorophyll-a concentration (CHL-a) and the anomaly (CHL-a $_{anom}$) using
satellite derived monthly product in 9 km resolution (Hu et al., 2012), mixed layer depth
(MLD) and sea surface height (SSH) and the anomalies (MLD$_{anom}$ and SSH$_{anom}$) using
the ECCO2 cube92 daily product (Menemenlis et al., 2008), W velocity of ocean
currents (W$_{vel}$) at 5, 65, 105 and 195 m depth using the ECCO2 cube92 3-day product
(Menemenlis et al., 2008), dry air mixing ratio of atmospheric CO$_2$ (xCO$_2$) and the
anomaly (xCO$_2$ $_{anom}$) from the GLOBAL VIEW marine boundary layer product
(GLOBALVIEW-CO2, 2011), sea ice area fraction using the monthly product from
ECMWF ERA Interim(Dee et al., 2011), 10 meters wind speed using the monthly
product from ECMWF ERA Interim (Dee et al., 2011), bathymetry from ETOPO2
(Commerce et al., 2006) , year and month (represented by 1-12), the total number of
months since January 1992 (N$_{mon}$), the sine of latitude and the sine and cosine of
longitude (sLat, sLon and cLon). In addition, 8 parameters which were only used in
similar previous research focused on other parameters, or were possibly related to the
driver of surface ocean $p$CO$_2$ and its variability, were selected to be tested. These
parameters included nitrate, phosphate, silicate and dissolved oxygen (DO) using the
monthly climatology product from WOA18 (Garcia et al., 2019a, b), sea level pressure
(SLP) and surface pressure from the ECMWF ERA Interim (Dee et al., 2011), the
Oceanic Nino Index (ONI) (Huang et al., 2017), the Southern Hemisphere Annular
Mode Index (SAM) (Marshall, G. J., 2003).
**2.2 Biogeochemical provinces defined by the Self-Organizing Map**
For applying different combination of indicators in regions based on the differences
in the dominated drivers of $p$CO$_2$ and its variability, the global ocean was divided into
a set of biogeochemical provinces using a Self-Organizing Map (SOM) method. The
monthly climatology of temperature, salinity, nitrate, phosphate, silicate, and dissolved
oxygen were put into a 3-by-4 size SOM networks to generate 12 biogeochemical
provinces, where the monthly climatology data in all 12 months were put into one SOM
network to generate one discrete set of biogeochemical provinces. Then the discrete
small "island" provinces and provinces lack of SOCAT $p$CO$_2$ data were merged into the





nearest dominated province, and the provinces covering areas separated by land were
further divided artificially. The final version includes total 11 biogeochemical provinces.
In this study the coastal area was not involved and the boundary was defined as 200m
depth. In addition, the $p$CO$_2$ mapping based on SOM defined provinces tend to be less
smooth near the border of different biogeochemical provinces, with obvious border line
appearing. However, applying of different predictors may make this problem worse. To
obtain a smoother distribution, we defined that the grid within 5 1°×1° grids of province
borders belong to all provinces adjacent to the nearest province border. Samples in these
grids were involved in the FFNN training process of multiple provinces, but only
counted once in the validation.

**2.3 Stepwise FFNN algorithm**

For finding better combination of $p$CO$_2$ predicators, a stepwise FFNN algorithm
was constructed. We used the idea of the multiple linear stepwise regression, replacing
the linear regression part by a Feed-forward neural networks (FFNN). The mean
absolute error (MAE) difference that before and after adding or removing one indicator
in the input of FFNN was used to estimate the performance of each indicator in the
FFNN predicating. Although the root mean square error (RMSE) was widely used for
the validation of machine learning methods. Compared to the MAE, the RMSE was
more sensitive to a few extreme samples, which were generally deviated far from the
FFNN predicting values, resulting in a huge discrepancy between the FFNN outputs
and $p$CO$_2$ observations sometimes up to hundreds of μatm. A higher weight may be put
on these few extreme samples than other samples in the predicator selection if the
performance of each indicator was estimated by RMSE in the stepwise FFNN algorithm.
To avoid the higher weight on these few extreme samples, the MAE was used instead
in the stepwise FFNN algorithm. The basic principle of the stepwise FFNN algorithm
was adding each indicator from a set of indicators into the inputs of FFNN and
removing each redundant indicator from the inputs successively to reduce the MAE
between the FFNN outputs and SOCAT $p$CO$_2$ values in the fastest way, until no
decrease in the MAE appearing (Fig. 1), where the indicator having no contribution to
reduce the prediction error was considered as redundant.



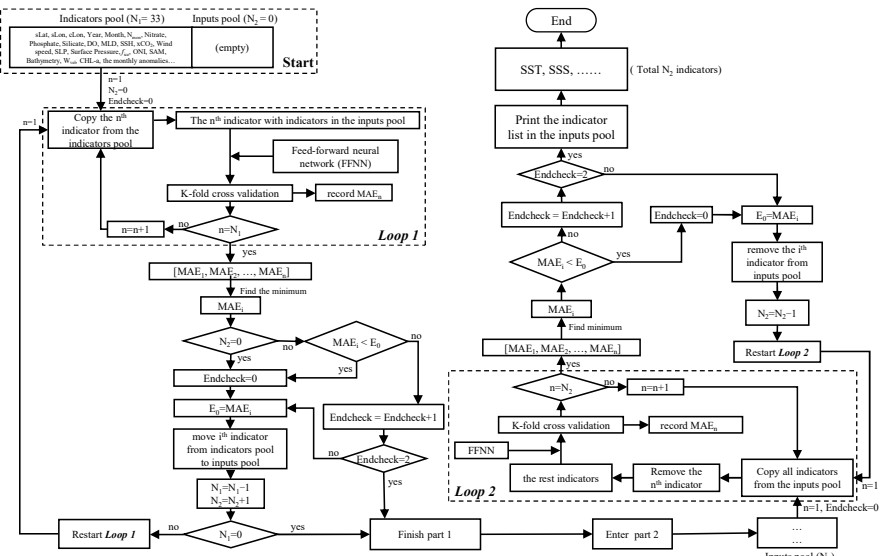

Figure 1. the procedure of stepwise FFNN algorithm
In the beginning of the stepwise FFNN algorithm, all available indicators were put
into a matrix, referred to as indicators pool, where each of rows represents one indicator
and each of columns represents one SOCAT sample. In this work we collected 33
parameters for test, that is, the indicators pool matrix has 33 rows. Meanwhile a matrix,
referred to as inputs pool, was set up to storage indicators with good performance,
where good performance means that adding these indicators as predicators can
significantly decrease the MAE between SOCAT $p\mathrm{CO_2}$ measurements and FFNN $p\mathrm{CO_2}$
predictions. Then a loop of K-fold validation test run out to calculated the MAE that
predicting $p\mathrm{CO_2}$ by each one indicator in the indicators pool in the first step (loop 1 in
the Fig. 1). Thus total 33 MAE values were obtained and the minimum was recorded as
$E_0$. The indicator that corresponds to the minimum of all MAE values was moved from
the indicators pool to the inputs pool. After that the loop 1 restarted, i.e., the second step
started with one indicator removed to the inputs pool and the rest 32 indicators waiting
to be tested. Then 32 MAE values of predicting $p\mathrm{CO_2}$ by each one of the rest indicators
in the indicators pool with the addition of all indicators in the inputs pool were
calculated out. If the MAE in the lowest situation, represented by the $\mathrm{MAE_i}$, decreased
compared to the $E_0$, the $i^{\mathrm{th}}$ indicator was considered as a good indicator and was moved
from the indicators pool to the inputs pool as well. Then the value of $E_0$ was replaced
by the $\mathrm{MAE_i}$. This part was repeated that the good indicators were selected out in one-
by-one step and moved to the inputs pool in the way that the $E_0$ decreases in the fastest
way, until no indicator was left in the indicators pool or no decrease can be found no



matter which indicator was added in the next two steps. At this time the part 1 of
stepwise FFNN algorithm finished, and all indicators left in the indicators pool were
considered redundant. The loop K-fold validation in the second part run out in a
opposite way that the MAE was calculated with the indicators were removed from the
inputs pool one by one in the way that the $E_0$ decreases the fastest (loop 2 in Fig. 1).
The second part was aimed to remove the indicator that can be represented by other
indicators in the inputs pool, and finished in the similar condition that no significant
decrease can be found no matter which indicator was removed in the next two steps.
The FFNN is composed of four main parts, which are namely input, hidden,
summation and output layer (Fig.2). The input layer is designed to pass the inputs to
the hidden layer and the number of neurons is equal to the dimensions of the input
matrix $p$. The hidden layer includes 25 neurons in the FFNN model, with the tan-
sigmoid function as the transfer function. The input $p$ is multiplied by a matrix of
weights ($w_1$ in Fig. 2) and the inner product between the result and a bias matrix ($b_1$ in
Fig. 2) is calculated as the input of the transfer function in the first hidden layer. In the
summation layer, the transfer function $f_2$ is a pure linear function. The output of the
hidden layer is multiplied by another matrix of weights and summed. All bias and
weights matrixes were randomly assigned in the beginning of FFNN training. Here we
set one constant random number stream in the MATLAB, thus the way that the bias and
weights matrixes randomly assigned were steady, avoiding the appearance of
inconsistent results when algorithm repeats.

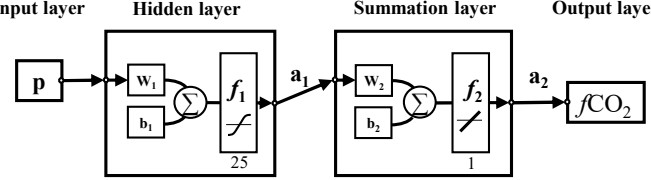


Figure 2. The structure of feed-forward neural network. **p**: input matrix; **w**: weighted matrix; **b**: bias
matrix; ∑: sum; $f_1$: tan-sigmoid transfer function; $f_2$: pure linear function; **a**: output matrix.
**2.4 $pCO_2$ product**
Dataset of parameters except CHL-a start since 1992 or earlier, while CHL-a data
ranges from 2002 to present. In each one of the provinces, the stepwise FFNN algorithm
was run out once first based on all samples covered by CHL-a data, then the algorithm
was run out secondly based on samples and all indicators except CHL-a and CHL-a $_{anom}$
in the year that CHL-a gridded data was not available. The $pCO_2$ mapping in the year
that CHL-a gridded data was not available was carried out based on the predicators
selected in the second run. Although the performance may improve with the number of





neurons increasing, the influence of number of neurons on the performance of FFNN
$pCO_2$ prediction remains unclear. To further decrease the predicating error between
FFNN outputs and SOCAT measurements, the number of neurons was improved by an
error test in each province. The number of neurons increased from 10 to 70 and the
corresponding MAE values of each size were record, and then the number of neurons
with lowest MAE was applied. This test avoided the appearance of insufficient learning
capacity for complex nonlinear relationship due to too few neurons and overfitting
problem due to too many neurons. Finally, based on the indicators selected by the
stepwise FFNN algorithm and improved FFNN size, a monthly global 1°×1° surface
ocean $pCO_2$ product from 1992 to 2019 was constructed.
**2.5 Validation**

To better estimate the predicating error of FFNN, the MAE and additionally the
RMSE which was widely used in previous researches, were calculated using a K-fold
cross validation method. To avoid overfitting caused by a lack of independence between
the training samples and testing samples, the SOCAT samples were put in chronological
order and then divided into group of years (Table 1). In this paper, the value of K was
set as 4. Thus, among every 4 neighboring years, three group samples were used for
training FFNN model and the rest one was used for testing. Total 4 iterations were
carried out, where testing year changed in each iteration. After 4 iterations finished, all
samples have been used for testing only once, and the MAE and RMSE between FFNN
output and the testing samples was calculated. The performance of the predicator
selection algorithm was estimated by comparing the MAE and RMSE result of the
FFNN based on stepwise selected indicators with the result based on indicators used in
previous researches in each biogeochemical province (Table 2). All validation groups
were applied with same FFNN and same samples from SOCAT, with the only
differences in predicators. Same K-fold validation procedure was applied for three
validation groups based on different $pCO_2$ predicators. Thus, three results were
generated to estimate whether the stepwise FFNN algorithm can effectively find better
combination of $pCO_2$ predictors. Finally the $pCO_2$ data generated in all validation
groups were further compared with the independent observations from the Hawaii
Ocean Time-series (HOT) (Dore et al., 2009), Bermuda Atlantic Time-series Study
(BATS) (Bates, 2007) and The European Station for Time Series in the Ocean Canary
Islands (ESTOC) (González-Dávila and Santana-Casiano, 2009) time series station.






Table 1. The procedure of K-fold validation.

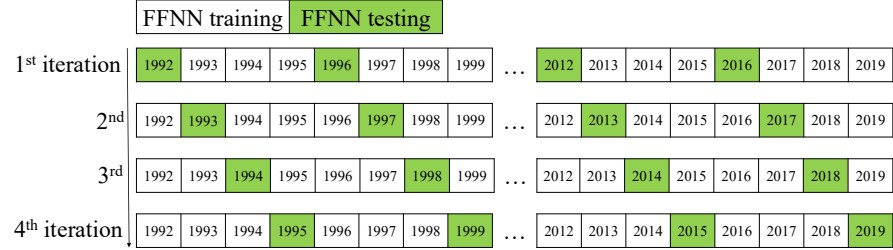

Table 2. validation group

| Validation group | Predictor |
|---|---|
| FFNN1 | Indicators selected by stepwise FFNN algorithm |
| FFNN2 | SST, SSS, $\log_{10}$(MLD), CHL-a, xCO$_2$, SST$_{anom}$, SSS$_{anom}$, xCO$_{2\ anom}$, CHL-a $_{anom}$, $\log_{10}$(MLD) $_{anom}$ (Landschuetzer et al., 2014) |
| FFNN3 | SST, SSS, SSH, MLD, xCO$_2$, CHL-a, SSS$_{anom}$, SST$_{anom}$, SSH$_{anom}$, CHL-a $_{anom}$, MLD$_{anom}$, xCO$_{2\ anom}$, sLat, sLon, cLon (Denvil-Sommer et al., 2019) |

## 3 Results and discussion

### 3.1 Biogeochemical provinces and corresponding predictors of $p$CO$_2$

11 biogeochemical provinces generated from the SOM method after the separated
small 'island' was removed and the province separated by lands was divided manually
(Fig. 3). The results of the stepwise FFNN algorithm in each province were shown in
the Table 3. The indicators were listed in the order that the stepwise FFNN algorithm
printed recommended predicators out. The indicator printed earlier was relatively more
recommended and played an important role in the prediction of $p$CO$_2$ based on FFNN.
Applying of these indicators as the predicators of surface ocean $p$CO$_2$ effectively
decreased the predicating error between the FFNN outputs and $p$CO$_2$ values from
validation samples, thus it is reasonable to consider that these indicators were highly
related to the drivers of $p$CO$_2$ and its variability. Indicators representing sampling
position were also listed as recommended predicators in some provinces, including
latitude, longitude and sampling time, suggesting that relatively steady spatial or
temporal variability pattern of surface ocean $p$CO$_2$ existed in these biogeochemical
provinces. For example, month was considered as a recommended predicator in most
provinces. Especially in the provinces covering the north Atlantic Ocean (P4 and P5),
the parameter month was relatively more recommended. While $p$CO$_2$ in these areas
regularly peaked and bottomed out in summer and winter (Takahashi et al., 2009;





Landschutzer et al., 2016; Landschützer et al., 2020). Similarly, latitude and the sine
and cosine of longitude were listed as recommended predicators of $pCO_2$ in most
provinces, suggesting an obvious spatial distribution pattern of $pCO_2$, which was not
learned sufficiently by the FFNN model from existing indicators and the indicators
related to spatial position were applied as supplementary.

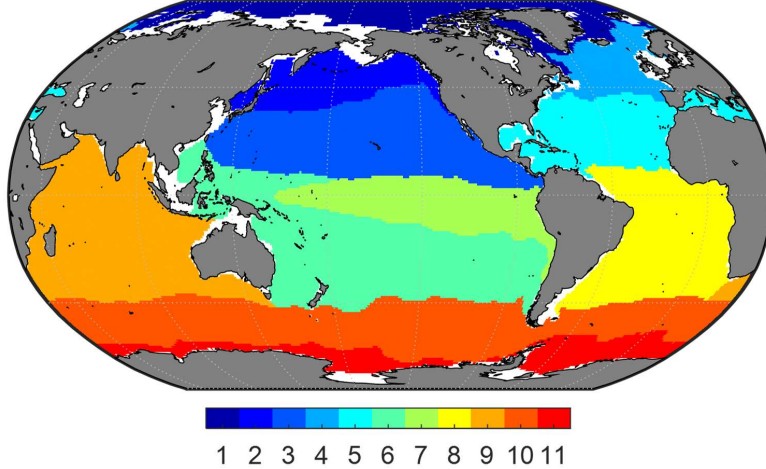

Figure 3. The map of biogeochemical provinces

As basic parameters highly related to the ocean environment, the temperature and
salinity was considered as parts of the most important predictors of surface ocean $pCO_2$,
and was applied in the $pCO_2$ prediction in almost all previous relating researches based
on various method (Jo et al., 2012; Signorini et al., 2013; Landschuetzer et al., 2014;
Marrec et al., 2015; Chen et al., 2016; Moussa et al., 2016; Chen et al., 2017; Laruelle
et al., 2017; Zeng et al., 2017; Chen et al., 2019; Denvil-Sommer et al., 2019). The
results of stepwise FFNN algorithm also supported this. Temperature was listed as a
recommended predictor in all biogeochemical provinces, suggesting that temperature
was the one of the most important drivers of $pCO_2$ and its variability in these provinces.
Similarly, the result of stepwise FFNN algorithm proved the importance of salinity in
the predication of $pCO_2$, which was also listed as a predicator in most provinces. In the
province P1 located in the Arctic, the silicate concentration and temperature were
considered as the most crucial predicator of $pCO_2$. The dry air mixing ratio of
atmospheric $CO_2$ (x$CO_2$) and the monthly anomaly of x$CO_2$ were also recommended
predicators in most of the biogeochemical provinces, suggesting that the exchange of
$CO_2$ across the sea-air interface was also an important driver of surface ocean $pCO_2$. As
a widely used predictor in the $pCO_2$ prediction, the chlorophyll-a concentration (CHL-

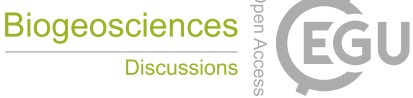



a) played an important role in fitting the influence of biological activities on $pCO_2$ in
previous researches (Landschuetzer et al., 2014; Zeng et al., 2017; Laruelle et al., 2017;
Denvil-Sommer et al., 2019). Especially in the Southern Ocean (province P10 and P11)
the CHL-a was listed as the most recommended predicator in the result of stepwise
FFNN algorithm. While in some other provinces (P1 and P5), the CHL-a were
considered redundant that no effective decrease of MAE between FFNN outputs and
$pCO_2$ measurements appeared when CHL-a data was used. Similar with the period that
CHL-a was not available (represented by the subscript 'b'), the phosphate, nitrate,
silicate or dissolved oxygen were recommended instead.

Table 3. Predicators in each biogeochemical province

| Province | Predictors |
|---|---|
| P1 | Silicate, SST, Wind speed, SSS, $\log_{10}(MLD)$, $SSS_{anom}$, sLat, month, $W_{vel}(65m)$, $\log_{10}(MLD)_{anom}$, $xCO_2$, cLon, Bathymetry, SSH |
| $P2_a$[*] | Nitrate, CHL-a, SSS, $xCO_2$, cLon, SST, $\log_{10}(MLD)$, sLon, sLat, month |
| $P2_b$[*] | Nitrate, $xCO_{2anom}$, sLon, SST, sLat, $\log_{10}(MLD)$, cLon, SSS, $SSH_{anom}$, DO, $W_{vel}(195m)$, Bathymetry, Silicate |
| $P3_a$ | $\log_{10}(MLD)$, $N_{mon}$, SSH, SST, sLon, sLat, SSS, Bathymetry, month, $\log_{10}(MLD)_{anom}$, cLon, Surface pressure, $W_{vel}(105m)$, CHL-a, DO, $SSH_{anom}$, $xCO_{2\ anom}$ |
| $P3_b$ | $\log_{10}(MLD)$, $xCO_2$, sLat, sLon, SST, Surface pressure, cLon, SSS, $W_{vel}(5m)$, $N_{mon}$, $\log_{10}(MLD)_{anom}$, month, Phosphate, $xCO_{2\ anom}$, $W_{vel}(105m)$ |
| $P4_a$ | month, sLat, cLon, SST, Year, CHL-a, DO, $SSS_{anom}$, $W_{vel}(195m)$, SSH, $\log_{10}(MLD)$, Bathymetry, SSS |
| $P4_b$[*] | month, $xCO_2$, DO, Wind speed, $\log_{10}(MLD)$, $W_{vel}(195m)$, sLon, Bathymetry, $W_{vel}(5m)$, SST, Phosphate, Year, $N_{mon}$ |
| P5 | month, Year, SST, sLon, sLat, SSS, $SST_{anom}$, SSH, Bathymetry, $W_{vel}(5m)$, cLon, $W_{vel}(65m)$, $\log_{10}(MLD)_{anom}$ |
| $P6_a$ | SST, sLon, $xCO_{2\ anom}$, sLat, SSS, month, Phosphate, CHL-a, $CHL\text{-}a_{anom}$, $W_{vel}(65m)$, $\log_{10}(MLD)$, $\log_{10}(MLD)_{anom}$, Nitrate, Bathymetry |
| $P6_b$ | $xCO_2$, sLat, SSS, SST, Phosphate, SLP, $xCO_{2\ anom}$, sLon, cLon, $W_{vel}(105m)$, $W_{vel}(65m)$, DO, Bathymetry, SSH, SAM |
| $P7_a$ | Nitrate, $xCO_2$, sLat, SSS, SST, cLon, $xCO_{2\ anom}$, $\log_{10}(MLD)$, sLon, CHL-a, Phosphate, $W_{vel}(5m)$, $W_{vel}(105m)$, $W_{vel}(195m)$ |
| $P7_b$ | SST, SSS, Year, sLat, month, cLon, SSH, Bathymetry, $W_{vel}(65m)$, $xCO_2$ |
| $P8_a$ | sLat, $xCO_{2\ anom}$, SSS, $\log_{10}(MLD)$, CHL-a, $SSH_{anom}$, $W_{vel}(195m)$, cLon, SST, $W_{vel}(65m)$, Bathymetry, Nitrate |



| Province | Predictors |
|---|---|
| P8$_b$ | SST, xCO$_2$, cLon, sLat, SSS, Silicate, SSH, log$_{10}$(MLD), sLon |
| P9$_a$ | SST, cLon, sLat, Nitrate, W$_{vel}$(65m), log$_{10}$(MLD), SLP, CHL-a, Year, log$_{10}$(MLD)$_{anom}$, SSH$_{anom}$ |
| P9$_b$ | SLP, month, sLon, xCO$_{2\ anom}$, SST, Silicate, W$_{vel}$(65m) |
| P10$_a$ | CHL-a, log$_{10}$(MLD), N$_{mon}$, SSS, SST, Bathymetry, SSH$_{anom}$, W$_{vel}$(5m), CHL-a$_{anom}$, xCO$_2$ |
| P10$_b$ | Wind speed, xCO$_{2\ anom}$, SSS, Phosphate, log$_{10}$(MLD), W$_{vel}$(65m), Bathymetry, SST, month |
| P11$_a$ | CHL-a, sLon, Bathymetry, SSS, SSH, SST, Nitrate, cLon, sLat |
| P11$_b$ | month, DO, SST, SSH, sLat, Nitrate, sLon, SSS, W$_{vel}$(195m), Silicate, SSHanom |

*: Due to insufficient coverage of CHL-a data in the polar areas and during the period before 2002. The *p*CO$_2$ data in the province that CHL-a or CHL-a $_{anom}$ was selected as predicators was divided into two periods. The period that CHL-a data available was represented by the subscript 'a', such as P2$_a$, including global grids from 2002 to 2019 except polar grids in winter. The period that CHL-a data not available was represented by the subscript 'b', such as P2$_b$, including global grids from 1992 to 2001 and additionally some polar grids in winter from 1992 to 2019.

**3.2 *p*CO₂ product**

Based on the predicators given by the stepwise FFNN algorithm in each biogeochemical province, a FFNN size (representing the number of neurons in the hidden layer) improving validation was applied to further decrease the predication error. The MAE values based on same samples and FFNN model with different number of neurons were calculated, then the number of neurons corresponding to the lowest MAE were applied (Fig. 4a). The MAE in most provinces tend to decrease first and then increase when the number of neurons in the hidden layer of FFNN model increased from 10 to 70. Based on the variation of MAE with the number of neurons in the FFNN hidden layer, the optimal FFNN size in each province was considered as the number of neurons when the MAE was lowest. The result and corresponding MAE were shown in Fig. 4b. The MAE and RMSE of global estimates between predicted *p*CO₂ and measurements from SOCAT v2020 further decreased to 11.32 and 17.99 µatm respectively after applying optimal FFNN size in each province.





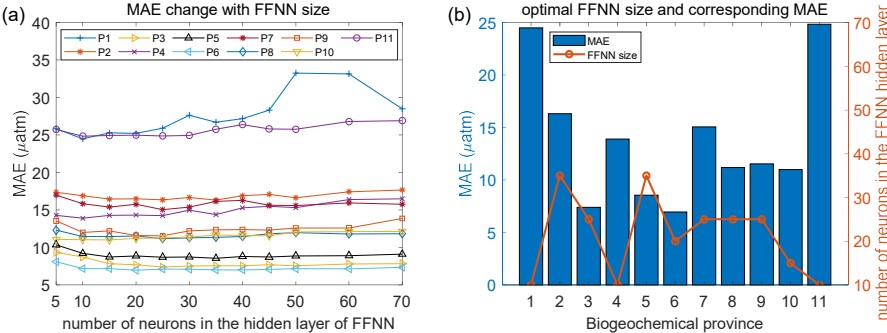

Figure 4. MAE of different FFNN size in each biogeochemical province.

Then the RMSE and mean residuals in each grid were calculated based the K-fold cross validation method. In most grids, the RMSE was lower than 10 µatm and the mean residuals was close to zero (Fig. 5). However, the prediction error in the north subpolar Pacific, the east equatorial Pacific and the Southern Ocean near the Antarctic continent was obviously higher than other areas. Distribution of mean residuals suggested that surface ocean $pCO_2$ in the Indian Ocean tend to be overestimated by the FFNN models. While in other regions the distribution of mean residuals was more discrete and no obvious pattern was found.

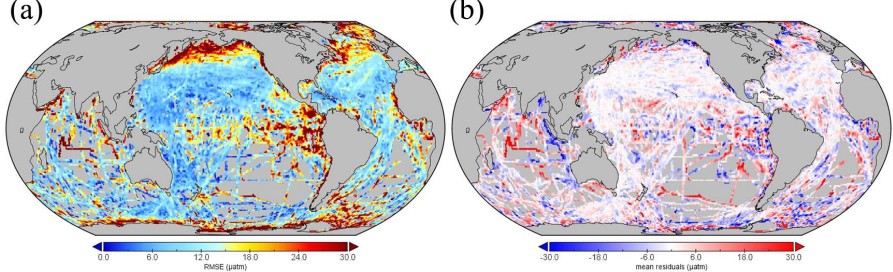

Figure 5. Global maps of (a) RMSE and (b) mean residuals between predicted $pCO_2$ and SOCAT observations

Based on stepwise FFNN algorithm and improved FFNN size in each province, a monthly 1°×1° grided surface ocean $pCO_2$ product from January 1992 to August 2019 was constructed. The interannual variability of global average $pCO_2$ was showed in the Fig. 6. The global open ocean average $pCO_2$ increased about 1.85 µatm per year from 1992 to 2019.

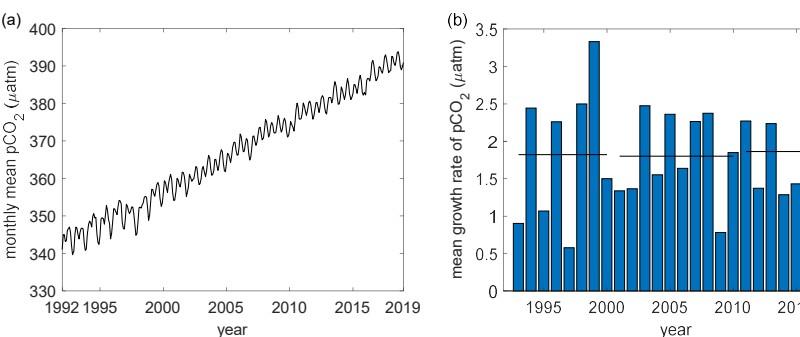

Figure 6. Interannual variability of global open-oceanic $pCO_2$ during 1992-2019. (a): global
monthly mean $pCO_2$, (b): growth rate of global monthly mean $pCO_2$
**3.3 Validation of the stepwise FFNN algorithm based on SOCAT samples**
Validation based on the K-fold cross validation method suggested that most FFNN
outputs were quite close to the $pCO_2$ values from SOCAT v2020 samples (Fig. 7).
Comparing the results based on different combination of predicators, the results of
FFNN1 (based on stepwise FFNN algorithm,this paper) and FFNN3 (based on 15
predicators from Denvil-Sommer, et al. 2019) were obviously more precise than that of
FFNN2 (based on 10 predicators from Landschuetzer, et al. 2014). Where the plots in
the result of FFNN1 was most concentrated along the y=x line, suggesting extremely
close FFNN outputs with the measured $pCO_2$ values from SOCAT, with the RMSE of
17.99 µatm in the global open oceans. The RMSE of FFNN1 was lower than that of
FFNN2 (22.95 µatm) and FFNN3 (19.17 µatm).

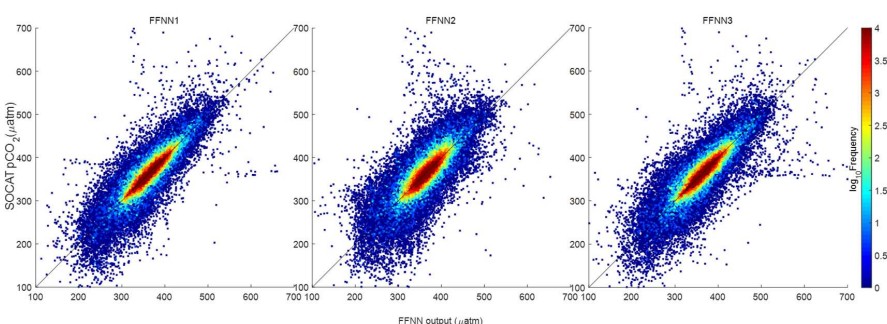

Figure 7. Comparing of FFNN predicted $pCO_2$ with SOCAT $pCO_2$. FFNN1 was based on predictors
selected by the stepwise-FFNN algorithm. FFNN2 and FFNN3 were based on predictors from
Landschuetzer et al., 2014 and Denvil-Sommer et al., 2019 respectively.
For specific comparison of accuracy in each province, the MAE of FFNN1 was
lower in most provinces (Table. 4), except the relatively close results between the
FFNN1 and FFNN3 in parts of provinces. Where the MAE of FFNN1 in the province





P9 was significantly lower than that of the other validation groups, suggesting a better
combination of predicators highly related to the drivers of surface ocean $pCO_2$ and its
variability in the Indian Ocean. Compared with predicators of FFNN2 and FFNN3, the
predicators of FFNN1 added surface pressure and W velocity of ocean currents, and
abandoned the monthly anomalies of other indicators in the province P9. The low
relevance between part of the monthly anomalies, such as $SSS_{anom}$ and $SSH_{anom}$, may
be responsible for significant lower MAE of FFNN1. Adding redundant indicators may
cause misleading in the learning of FFNN model on the contrary. The MAE and RMSE
difference between FFNN1 and FFNN3 in some provinces were relatively small,
because predicators used in both FFNN1 and FFNN3 were related to main drivers of
$pCO_2$, such as CHL-a, xCO2 and MLD.
Table 4. Performance of the $pCO_2$ prediction based on different predicators

| Province | FFNN size | MAE (µatm) | | | RMSE (µatm) | | |
|---|---|---|---|---|---|---|---|
| | | FFNN1 | FFNN2 | FFNN3 | FFNN1 | FFNN2 | FFNN3 |
| P1 (9856) | 10 | 24.50 | 32.32 | 26.87 | 32.27 | 43.68 | 35.08 |
| P2 (30516) | 35 | 16.32 | 20.63 | 16.67 | 24.32 | 29.87 | 25.03 |
| P3 (56367) | 25 | 7.39 | 12.16 | 7.95 | 11.33 | 17.75 | 11.88 |
| P4 (29595) | 10 | 13.89 | 16.91 | 14.73 | 21.06 | 24.29 | 22.27 |
| P5 (45358) | 35 | 8.55 | 12.28 | 9.00 | 12.80 | 17.86 | 13.72 |
| P6 (31803) | 20 | 6.96 | 9.94 | 7.24 | 9.86 | 14.64 | 11.00 |
| P7 (11233) | 25 | 15.05 | 19.55 | 15.49 | 20.98 | 27.61 | 21.10 |
| P8 (10259) | 25 | 11.19 | 15.07 | 12.43 | 17.10 | 20.87 | 17.66 |
| P9 (7440) | 25 | 11.54 | 13.78 | 15.49 | 17.15 | 22.89 | 28.29 |
| P10 (21206) | 15 | 11.00 | 11.76 | 12.14 | 16.61 | 17.22 | 17.66 |
| P11 (10683) | 10 | 24.84 | 29.26 | 25.74 | 34.73 | 40.42 | 35.22 |
| Global (264316) | | 11.32 | 15.08 | 12.06 | 17.99 | 22.95 | 19.17 |

(FFNN1 was based on predicators selected by the stepwise-FFNN algorithm. FFNN2 and FFNN3
were based on predicators from Landschuetzer et al., 2014 and Denvil-Sommer et al., 2019
respectively.)

**3.4 Validation based on independent observations**
The FFNN outputs based on different combination of predicators were compared
with independent observations from the Ocean Time-series (HOT) (Dore et al., 2009),

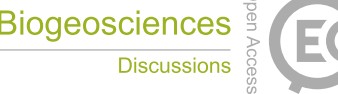

Bermuda Atlantic Time-series Study (BATS) (Bates, 2007) and The European Station
for Time Series in the Ocean Canary Islands (ESTOC) (González-Dávila and Santana-
Casiano, 2009) (Fig. 8). The interannual variability and seasonal pattern of $p$CO$_2$ in the
grids the HOT station located from different validation groups were similar and close
to the observations from the HOT, which was located in the province P3. From 1992 to
2019, the RMSE between FFNN1 outputs and HOT observations was only 9.29 μatm,
lower than the 10.85 μatm of FFNN2 and the 10.70 μatm of FFNN3.The monthly mean
$p$CO$_2$ of FFNN2 during winter was obviously lower than the HOT observations and
$p$CO$_2$ values of other validation groups, while the FFNN1 and FFNN3 outputs were
closer to the HOT observations. MAE between predicted $p$CO$_2$ and HOT observations
were also lower in the validation group FFNN1, which was only 7.17 μatm, compared
to the 8.61 μatm of FFNN2 and the 8.44 μatm of FFNN3. Higher bias generated in the
winter bottom and summer peak, which was showed more obviously in the monthly
average of $p$CO$_2$ (Fig. 8b). Compared with other validation groups, the result of FFNN1
was closer to the monthly average values of the HOT observations. Same conclusion
can be obtained in the ESTOC and BATS station located in the province P5. The RMSE
between FFNN1 outputs and independent observations were 13.03 μatm in the BATS
station and 11.35 μatm in the ESTOC station, lower than that of other validation groups.
The RMSE between FFNN2 outputs and independent observations was 16.15 μatm in
the BATS station and 14.51 μatm in the ESTOC station. For the group FFNN3, the
RMSE was 13.09 μatm in the BATS station and 13.01 μatm in the ESTOC station. All
results were extremely close to the independent observations, but the RMSE and MAE
of FFNN1 were lower. Similar with the situation in the HOT station, the FFNN1 was
most close and the FFNN3 second. Based on the better performance of FFNN1, in
which the predicators selected by stepwise FFNN algorithm were used, we may
conclude that the stepwise FFNN algorithm can effectively find better combination of
predictors to fit the diver of surface ocean $p$CO$_2$ and obtained lower error.



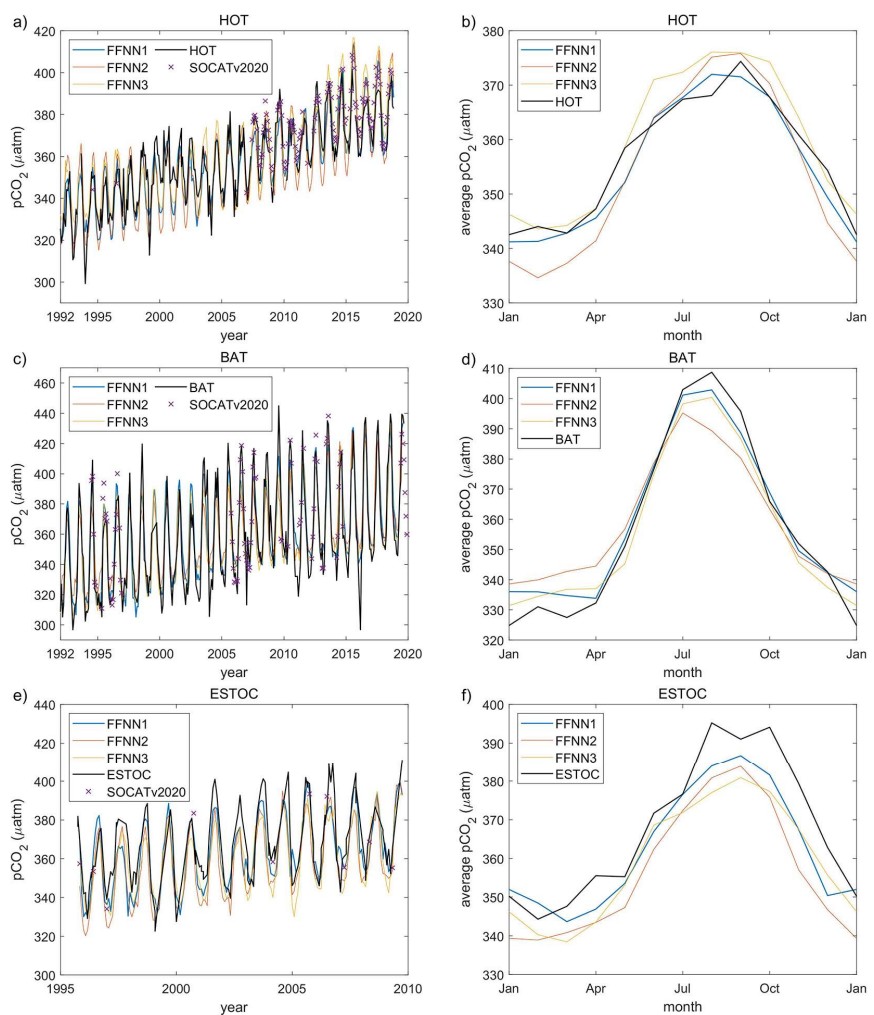

Figure 8. Validation based on independent observation from time series stations. a) and b): the Hawaii Ocean Time-series (HOT) (Dore et al., 2009); c) and d): the Bermuda Atlantic Time-series Study (BATS) (Bates, 2007); e) and f): the European Station for Time Series in the Ocean Canary Islands (ESTOC) (González-Dávila and Santana-Casiano, 2009) time series station. FFNN1 was based on predictors selected by the stepwise-FFNN algorithm. FFNN2 and FFNN3 were based on predictors from Landschuetzer et al., 2014 and Denvil-Sommer et al., 2019 respectively. SOCATv2020 represents the monthly mean $p$CO$_2$ of SOCAT observations in the corresponding grids of each time series station.

**3.5 Climatological spatial distribution**

The climatological average distribution of $p$CO$_2$ suggested a significant spatial variability (Fig. 9), which is consistent with the average distribution of SOCAT



observations. In the Pacific Ocean, the high $pCO_2$ areas showed by the stepwise-FFNN
product (Fig. 9b), including the equatorial areas, east temperate areas and north
subpolar areas, were highly consistent with the SOCAT datasets (Fig. 9a). Similarly, the
distribution of $pCO_2$ in the Atlantic Ocean and the Indian Ocean was also close.
However, the stepwise-FFNN product suggested lower $pCO_2$ average values in the
Arctic and higher values in the Southern Ocean near the Antarctic continent. Compared
with previous climatology product (Landschützer et al., 2020), the global distribution
pattern of surface ocean $pCO_2$ was basically well consistent. Inconsistent spatial
distribution also existed in the Arctic and parts of the Southern Ocean near the Antarctic
continent. The differences between stepwise-FFNN product and previous climatology
product may be caused by differences in methods or SOCAT dataset versions used.
While lower average values of the SOCAT dataset in the Southern Ocean may be caused
by the undersampling in winter. The global spatial distribution pattern of the stepwise
FFNN $pCO_2$ product was basically well consistent with previous climatology product
and SOCAT dataset, suggesting that $pCO_2$ predicting based on regional different
predictors selected by the stepwise FFNN algorithm was credible.




Figure 9. Comparison between long term average of a): SOCAT v2020 dataset, b): the stepwise
FFNN $p$CO$_2$ product and c): previous climatology product adapted from Landschützer et al., 2020.



## 4. Conclusions

A stepwise FFNN algorithm was constructed to decreasing the predicating error in the surface ocean $p\mathrm{CO_2}$ mapping by finding better combinations of $p\mathrm{CO_2}$ predicators in each biogeochemical province defined by SOM method. Comparing with the performance of FFNN based on predicators same with previous researches, the RMSE decreased when using predicators selected by the stepwise FFNN algorithm in all provinces, suggesting that the stepwise FFNN algorithm was capable to find better combination of predicators. In addition, validation based on independent observations from HOT, BATS and ESTOC time series stations also proved the better performance of FFNN based on predicators selected by the stepwise FFNN algorithm. We further decreased the MAE and RMSE of global estimates to 11.32 and 17.99 μatm by improving the number of neurons in the hidden layer of FFNN. Then a monthly 1°×1° gridded global open-oceanic surface ocean $p\mathrm{CO_2}$ product from January 1992 to August 2019 was constructed, based improved FFNN size and the predicators selected by stepwise FFNN algorithm. In this study, regional specific combination of predicators was first applied in the global surface ocean $p\mathrm{CO_2}$ mapping. The result of the stepwise FFNN algorithm was also capable for analyzes of driving based on the ranking of relative importance of each predicator. The more important predicator, which played a more important role in decreasing the predicting error, will be selected earlier and listed at the front of the recommended predicator list. In the future work, the stepwise FFNN algorithm will be attempted in the mapping of other parameters, such as total alkalinity and pH, to provide more sufficient data support for studies on ocean acidification and carbon cycling.

**Code and data availability**

The stepwise FFNN algorithm (as a .m file for MATLAB) and the global 1°×1° gridded surface ocean $p\mathrm{CO_2}$ product since from January 1992 to August 2019 (as a NetCDF file) generated during this study is available from the Institute of Oceanology of the Chinese Academy of Sciences Marine Science Data Center at http://dx.doi.org/10.12157/iocas.2021.0022

**Author contribution**

Ma Jun, Yuan Huamao and Duan Liqin collected the dataset of $p\mathrm{CO_2}$ predicators, and Qu baoxiao and Wang Yanjun was contributed in the synthesis of datasets. Zhong Guorong, Li Xuegang and Song Jinming designed the predicator selection algorithm and performed the reconstruction of $p\mathrm{CO_2}$ product. Wang Fan, Zhang Bin, Sun Xiaoxia, Zhang Wuchang, and Wang Zhenyan were contributed in the further improving. Zhong



Guorong prepared the manuscript with contributions from all co-authors.

**Competing interests**

The authors declare that they have no conflict of interest.

**Acknowledgement**

This work was supported by The National Key Research and Development
Program of China (No. 2017YFA0603204), the Strategic Priority Research Program of
the Chinese Academy of Sciences (No. XDA19060401), National Natural Science
Foundation of China (No. 91958103 and No. 42176200), and Natural Science
Foundation of Shandong Province (ZR2020YQ28). We thank SOCAT for sharing the
$f$CO$_2$ observation data. The Surface Ocean CO$_2$ Atlas (SOCAT) is an international effort,
endorsed by the International Ocean Carbon Coordination Project (IOCCP), the Surface
Ocean Lower Atmosphere Study (SOLAS) and the Integrated Marine Biosphere
Research (IMBeR) program, to deliver a uniformly quality-controlled surface ocean
CO$_2$ database. The many researchers and funding agencies responsible for the collection
of data and quality control are thanked for their contributions to SOCAT.

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

Methods of Seawater Analysis, 3rd edn., 1999.

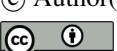



Landschuetzer, P., Gruber, N., Bakker, D. C. E., and Schuster, U.: Recent variability
of the global ocean carbon sink, Glob. Biogeochem. Cycle, 28, 927-949,
10.1002/2014gb004853, 2014.
Landschutzer, P., Gruber, N., Bakker, D. C. E., Schuster, U., Nakaoka, S., Payne, M.
R., Sasse, T. P., and Zeng, J.: A neural network-based estimate of the seasonal to
inter-annual variability of the Atlantic Ocean carbon sink, Biogeosciences, 10,
7793-7815, 10.5194/bg-10-7793-2013, 2013.
Landschutzer, P., Gruber, N., Haumann, A., Rodenbeck, C., Bakker, D. C. E., van
Heuven, S., Hoppema, M., Metzl, N., Sweeney, C., Takahashi, T., Tilbrook, B., and
Wanninkhof, R.: The reinvigoration of the Southern Ocean carbon sink, Science,
349, 1221-1224, 10.1126/science.aab2620, 2015.
Landschutzer, P., Gruber, N., and Bakker, D. C. E.: Decadal variations and trends of
the global ocean carbon sink, Glob. Biogeochem. Cycle, 30, 1396-1417,
10.1002/2015gb005359, 2016.
Landschützer, P., Laruelle, G. G., Roobaert, A., and Regnier, P.: A uniform $pCO_2$
climatology combining open and coastal oceans, Earth Syst. Sci. Data Discuss.,
2020, 1-30, 10.5194/essd-2020-90, 2020.
Laruelle, G. G., Landschutzer, P., Gruber, N., Tison, J. L., Delille, B., and Regnier, P.:
Global high-resolution monthly $pCO_2$ climatology for the coastal ocean derived
from neural network interpolation, Biogeosciences, 14, 4545-4561, 10.5194/bg-14-
4545-2017, 2017.
Marrec, P., Cariou, T., Mace, E., Morin, P., Salt, L. A., Vernet, M., Taylor, B., Paxman,
632        K., and Bozec, Y.: Dynamics of air-sea $CO_2$ fluxes in the northwestern European
shelf based on voluntary observing ship and satellite observations, Biogeosciences,
12, 5371-5391, 10.5194/bg-12-5371-2015, 2015.
Marshall G J.: Trends in the Southern Annular Mode from observations and
reanalyses, Journal of climate, 16, 10.1175/1520-
0442(2003)016<4134:TITSAM>2.0.CO;2, 4134-4143, 2003.
Menemenlis, D., Campin, J.-M., Heimbach, P., Hill, C., Lee, T., Nguyen, A.,
Schodlok, M., and Zhang, H.: ECCO2: High Resolution Global Ocean and Sea Ice
Data Synthesis, Mercator Ocean Quarterly Newsletter, 2008.
Moussa, H., Benallal, M. A., Goyet, C., and Lefevre, N.: Satellite-derived $CO_2$
fugacity in surface seawater of the tropical Atlantic Ocean using a feedforward
neural network, Int J Remote Sens, 37, 580-598, 10.1080/01431161.2015.1131872,
2016.
Nakaoka, S., Telszewski, M., Nojiri, Y., Yasunaka, S., Miyazaki, C., Mukai, H., and
Usui, N.: Estimating temporal and spatial variation of ocean surface $pCO_2$ in the
North Pacific using a self-organizing map neural network technique,
Biogeosciences, 10, 6093-6106, 10.5194/bg-10-6093-2013, 2013.
Regnier, P., Friedlingstein, P., Ciais, P., Mackenzie, F. T., Gruber, N., Janssens, I. A.,
Laruelle, G. G., Lauerwald, R., Luyssaert, S., Andersson, A. J., Arndt, S., Arnosti,
C., Borges, A. V., Dale, A. W., Gallego-Sala, A., Godderis, Y., Goossens, N.,
Hartmann, J., Heinze, C., Ilyina, T., Joos, F., LaRowe, D. E., Leifeld, J., Meysman,
F. J. R., Munhoven, G., Raymond, P. A., Spahni, R., Suntharalingam, P., and





Thullner, M.: Anthropogenic perturbation of the carbon fluxes from land to ocean, Nature Geoscience, 6, 597-607, 10.1038/Ngeo1830, 2013.

Sabine, C. L., Feely, R. A., Gruber, N., Key, R. M., Lee, K., Bullister, J. L., Wanninkhof, R., Wong, C. S., Wallace, D. W. R., Tilbrook, B., Millero, F. J., Peng, T. H., Kozyr, A., Ono, T., and Rios, A. F.: The oceanic sink for anthropogenic $CO_2$, Science, 305, 367-371, DOI 10.1126/science.1097403, 2004.

Sarma, V. V. S. S.: Monthly variability in surface $pCO_2$ and net air-sea $CO_2$ flux in the Arabian Sea, Journal of Geophysical Research-Oceans, 108, Artn 3255, 10.1029/2001jc001062, 2003.

Sarma, V. V. S. S., Saino, T., Sasaoka, K., Nojiri, Y., Ono, T., Ishii, M., Inoue, H. Y., and Matsumoto, K.: Basin-scale $pCO_2$ distribution using satellite sea surface temperature, Chl-a, and climatological salinity in the North Pacific in spring and summer, Glob. Biogeochem. Cycle, 20, Artn Gb3005, 10.1029/2005gb002594, 2006.

Schuster, U., McKinley, G. A., Bates, N., Chevallier, F., Doney, S. C., Fay, A. R., Gonzalez-Davila, M., Gruber, N., Jones, S., Krijnen, J., Landschuetzer, P., Lefevre, N., Manizza, M., Mathis, J., Metzl, N., Olsen, A., Rios, A. F., Roedenbeck, C., Santana-Casiano, J. M., Takahashi, T., Wanninkhof, R., and Watson, A. J.: An assessment of the Atlantic and Arctic sea-air CO2 fluxes, 1990-2009, Biogeosciences, 10, 607-627, 10.5194/bg-10-607-2013, 2013.

Shadwick, E. H., Thomas, H., Comeau, A., Craig, S. E., Hunt, C. W., and Salisbury, J. E.: Air-Sea $CO_2$ fluxes on the Scotian Shelf: seasonal to multi-annual variability, Biogeosciences, 7, 3851-3867, 10.5194/bg-7-3851-2010, 2010.

Signorini, S. R., Mannino, A., Najjar, R. G., Friedrichs, M. A. M., Cai, W. J., Salisbury, J., Wang, Z. A., Thomas, H., and Shadwick, E.: Surface ocean $pCO_2$ seasonality and sea-air $CO_2$ flux estimates for the North American east coast, Journal of Geophysical Research-Oceans, 118, 5439-5460, 10.1002/jgrc.20369, 2013.

Takahashi, T., Sutherland, S. C., Feely, R. A., and Wanninkhof, R.: Decadal change of the surface water $pCO_2$ in the North Pacific: A synthesis of 35 years of observations, Journal of Geophysical Research-Oceans, 111, Artn C07s05, 10.1029/2005jc003074, 2006.

Takahashi, T., Sutherland, S. C., Wanninkhof, R., Sweeney, C., Feely, R. A., Chipman, D. W., Hales, B., Friederich, G., Chavez, F., Sabine, C., Watson, A., Bakker, D. C. E., Schuster, U., Metzl, N., Yoshikawa-Inoue, H., Ishii, M., Midorikawa, T., Nojiri, Y., Kortzinger, A., Steinhoff, T., Hoppema, M., Olafsson, J., Arnarson, T. S., Tilbrook, B., Johannessen, T., Olsen, A., Bellerby, R., Wong, C. S., Delille, B., Bates, N. R., and de Baar, H. J. W.: Climatological mean and decadal change in surface ocean $pCO_2$, and net sea-air $CO_2$ flux over the global oceans, Deep-Sea Research Part Ii-Topical Studies in Oceanography, 56, 554-577, 10.1016/j.dsr2.2008.12.009, 2009.

Telszewski, M., Chazottes, A., Schuster, U., Watson, A. J., Moulin, C., Bakker, D. C. E., Gonzalez-Davila, M., Johannessen, T., Kortzinger, A., Luger, H., Olsen, A., Omar, A., Padin, X. A., Rios, A. F., Steinhoff, T., Santana-Casiano, M., Wallace, D.

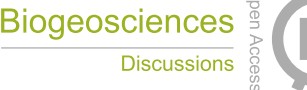

W. R., and Wanninkhof, R.: Estimating the monthly $pCO_2$ distribution in the North
Atlantic using a self-organizing neural network, Biogeosciences, 6, 1405-1421,
DOI 10.5194/bg-6-1405-2009, 2009.
Wang, Y., Li, X., Song, J., Zhong, G., and Zhang, B.: Carbon Sinks and Variations of
$pCO_2$ in the Southern Ocean from 1998 to 2018 Based on a Deep Learning
Approach, IEEE Journal of Selected Topics in Applied Earth Observations and
Remote Sensing, 2021.
Wanninkhof, R., Park, G. H., Takahashi, T., Sweeney, C., Feely, R., Nojiri, Y., Gruber,
N., Doney, S. C., McKinley, G. A., Lenton, A., Le Quere, C., Heinze, C.,
Schwinger, J., Graven, H., and Khatiwala, S.: Global ocean carbon uptake:
magnitude, variability and trends, Biogeosciences, 10, 1983-2000, 10.5194/bg-10-
1983-2013, 2013.
Watson, A. J., Schuster, U., Shutler, J. D., Holding, T., Ashton, I. G. C.,
Landschuetzer, P., Woolf, D. K., and Goddijn-Murphy, L.: Revised estimates of
ocean-atmosphere CO2 flux are consistent with ocean carbon inventory, Nature
Communications, 11, 10.1038/s41467-020-18203-3, 2020.
Weiss, R. F.: Carbon dioxide in water and seawater: the solubility of a non-ideal gas,
Marine Chemistry, 2, 203--215, 1974.
Zeng, J., Nojiri, Y., Landschützer, P., Telszewski, M., and Nakaoka, S.: A Global
Surface Ocean $f$CO$_2$ Climatology Based on a Feed-Forward Neural Network,
Journal of Atmospheric and Oceanic Technology, 31, 1838-1849, 10.1175/jtech-d-
13-00137.1, 2014.
Zeng, J. Y., Nojiri, Y., Nakaoka, S., Nakajima, H., and Shirai, T.: Surface ocean $CO_2$
in 1990-2011 modelled using a feed-forward neural network, Geosci Data J, 2, 47-
51, 10.1002/gdj3.26, 2015.
Zeng, J. Y., Matsunaga, T., Saigusa, N., Shirai, T., Nakaoka, S., and Tan, Z. H.:
Technical note: Evaluation of three machine learning models for surface ocean $CO_2$
mapping, Ocean Sci, 13, 303-313, 10.5194/os-13-303-2017, 2017.
Zhong, G., Li, X., Qu, B., Wang, Y., Yuan, H, and Song, J.: A General Regression
Neural Network approach to reconstruct global 1°×1° resolution sea surface $pCO_2$,
Acta Oceanol Sin, 10, 70-79, 10.3969/j.issn.0253-4193.2020.10.007, 2020.