# Peer review of "Reconstruction of global surface ocean pCO2 using"

_Biogeosciences, 2021_

## Referee Comment (RC1)

**Reconstruction of global surface ocean pCO2 using region-specific predictors based on a stepwise FFNN regression algorithm.**

The authors developed a new ML approach to reconstruct global surface ocean pCO2 that considers an impact of different predictors in different ocean regions. Based on Self-Organizing Map method authors defined 11 biogeochemical provinces. A stepwise FFNN regression algorithm was applied to each of these provinces to establish a set of predictors that are highly responsible for pCO2 variability in considered province. Based on selected predictors and analysis of FFNN size (number of neurons) a monthly 1°x1° surface ocean pCO2 product from 1992 to 2019 was constructed. The results show a good agreement with validation data and independent observations.

I found this work well-organized and easy to read. It was interesting to see new predictors (phosphate, nitrate, silicate, dissolved oxygen) and their role in pCO2 variability. The authors presented important results for the Indian Ocean where due to the lack of observations different methods show their disagreements.

Below, I listed several points that need to be clarified before publication.

Comments:
- Boundaries between provinces. In the text we can find "To obtain a smoother distribution, we defined that the grid within 5 1x1 grids of province borders belong to all provinces adjacent to the nearest province border. Samples in these grids were involved in the FFNN training process of multiple provinces, but only counted once in the validation." Please could you clarify what you mean by "only counted once in the validation"? Is only an output from one province used in the validation? If yes, how do you chose a province from which you take an output?
- Independent observations. Please could you provide geographical positions and period of stations used as independent observations?
- Set of selected predictors. In table 3 authors presented two sets for most of the regions that depends on the availability CHL-a data. Please could you present more explicitly that the final product is built on two FFNNs, one trained for the period 1992-2001 based on one predictors set and another – for 2002-2019 based on the second predictors set?
- More explicit figures' captions. Please provide more explicit figures' captions, period of presented results, or results averaged over xxxx-xxxx, what are horizontal lines in Fig.6b?
- Not correct conclusion. On page 15 lines 375-379 authors concluded that the difference between FNN1 and FFNN3 is relatively small, because predictors used in FFNN1 and FFNN3 were related to main drivers of pCO2, such as CHL-a, xCO2 and MLD. However, same drivers are used in FFNN2. Thus, it cannot explain why FFNN2 shows higher differences with observations.
- On page 18 line 430 authors said that the pattern of reconstructed pCO2 climatology was close to SOCAT in the Indian Ocean. I would say that it is not so close to mention it in this sentence.

Page 10 lines 298-300: For better structure of paragraph the sentence "In the province P1 located in the Arctic, the silicate concentration and temperature were considered as the most crucial predictor of pCO2." could be moved at the end of paragraph where authors mentioned the phosphate, nitrate, silicate, etc.

Page 13 lines 336-337: The sentence about results in the Indian Ocean can be removed if you put the Indian Ocean in the previous sentence, or please add "Also" at the beginning of the sentence dedicated to results in the Indian Ocean.

Page 16 lines 390-392: Please could you reformulate this sentence ("The interannual variability and seasonal pattern..") as it is difficult to read?

Page 18 lines 432-436: two sentence can be combined: "Compared with previous climatology product (Landschuster et al., 2020), the global distribution pattern of surface ocean pCO2 was basically well consistent: inconsistent spatial distribution also existed in the Arctic and parts of the Southern Ocean near the Antarctic continent."

Typo:
Page 2 line 41: "**s**urface ocean pCO2" should be replaced by "**S**urface ocean pCO2".
Page 9 line 261: "**v**alidation group" should be replaced by "**V**alidation group".
Page 13 line 332: "Based the K-fold" should be replaced by "based **on** the K-fold".
Page 20 line 459: "based improved FFNN size" should be replaced by "based **on** improved FFNN size".

---

## Referee Comment (RC2)

Review of the paper entitled "Reconstruction of global surface ocean pCO2 using region-specific predicators based on a stepwise FFNN regression algorithm" by Zhong et al.

General comments:
The stepwise FFNN looks like a innovative new approach to enhance the widely popular SOM-FFNN method. The stepwise method is tested using a very comprehensive list of predictors used elsewhere in the literature. The method building component looks thorough, congrats. The prediction of pCO2 based on region-specific predicators selected by the stepwise FFNN algorithm will be a valuable tool when moving to higher resolution, inside regional studies, or getting closer to shore. There are a number of grammatical errors that will need to be cleaned up by the authors or the journal team.

Specific comments:

**1 Introduction**

Line 66-82: Appreciate the list of previous works and predictor data used by each, provides justification for use in the stepwise FFNN. Would like to see one more sentence relating use of different predictors leading to varying marine sink estimates.

**2 Methodology**

**2.1 Data**

Line 106-122: Are all these products retrieved at the same resolution? Are they upscaled or downscaled at all to your needs?

Line 122: "In addition, 8 parameters…." Thanks for listing after this sentence. Which previous research used these as predictors in observation-based pCO2 estimates? List and provide citations like in the introduction. Or is the inclusion of these predictors' novel? If so, highlight that.

**2.2 Biogeochemical provinces defined by the Self-Organizing Map**

Line 134: These SOM predictors exclude most of the FFNN predictors discussed in the introduction. Was there a reason why "biological" predictors (i.e., nutrients and oxygen) are weighted so heavily in the SOM selection? Were more physical predictors (i.e., mixed layer depth, etc.) used in SOM testing to optimize provinces? Or similar to previous work (Landschützer), using published $p$CO$_2$ climatology as a predictor to determine provinces?

Line 135: Just out of curiosity, did the configuration (3-by-4 size) make much of a difference to SOM province distribution?

Line 141: The 200 m depth boundary is fairly close to shore. Is this a commonly used open oceanic / coastal ocean boundary? If so citations from other studies here.

Line 144: Unique way to address the SOM boundary issue. Cool.

**2.3 Stepwise FFNN algorithm**

Line 152-163: Clarify. Was the mean absolute error used for the internal MATLAB neural network performance loss function (in the training targets and validation targets steps used to end training), **also / or** as a means for evaluating the FFNN output pCO2 product to withheld data?

Line 172: "…referred to as indicators pool (Start in Fig. 1),…" Keep coming back to this Figure throughout if you can. Makes it easier to read and connect to the Figure.

Figure 1: More sub boxes (dotted lines), connected to text could also make it easier to follow. Steps between loop 1 and loop 2, steps between loop2 and end.

**2.4 pCO2 product**

Line 227: Reason why 10 and 70 are chosen? Is there a possibility that even in smaller provinces 10 neurons could lead to overfitting? The polar regions are set right at 10.

Line 231: Does this vary neuron number test really limit overfitting? Taking the lowest MAE from the internal train/validation split during FFNN training step just means it is likely replicating training data well and due to random split inside autocorrelated validation data this doesn't change much. Being clear in Line 152-163 about how / when MAE is used to evaluate could clear this up.

**2.5 Validation**

Line 237: Unique use of the K-fold cross validation method grouping by year.

**3 Results and discussion**

**3.1 Biogeochemical provinces and corresponding predictors of pCO2**

Table 3: Add more to the caption on the order of the predictors listed.

**3.2 pCO2 product**

Line 317: Does this mean you first went through the stepwise FFNN process using the same neuron number? Then when the best predictors where determined you used the varying neuron number test (from 10-70) to find the best neuron number? Then you used the K cross validation to test robustness? Clarify. Link to back to Figure 1 if you need to.

Figure 4: Still not sure on this test limiting overfitting. Looks like they all (except at the poles maybe because it is not well constrained…? As in your Table 4) just level out. Using the same FFNN predictors and the same targets how reproducible is this Figure? Or is it dependant on the initialization on that run?

Line 334: Good to state this up front. Other than these regions it does look good. However, if the goal from the introduction is get at the air-sea flux, how important are these regions for the global marine CO2 flux? Suggest in conclusions what could be done in the future to improve these regions?

Figure 6a: Would be nice to also have the atmospheric xCO2 product on this Figure for comparison.

**3.4 Validation based on independent observations**

Line 395: Nothing is obvious to every reader. Remove and clarify.

Line 442: "… was credible." Is consistent with and improves upon? Readers should want to believe in what you did! Got to sell it a bit!

**4. Conclusions**

Line 447-465: This needs a bit of a rework. Feels like recycled sentences from throughout. What should readers take away from your work? How can this approach be applied in other studies? Who benefits from this improvement? Where is more work needed (e.g., polar regions), how could improvements be made?

---

## Referee Comment (RC3)

This manuscript uses a stepwise feed-forward neural network (FFNN) to identify an optimal feature for the prediction of ocean pCO2. The authors first use a self-organizing map (SOM) to cluster the ocean into 12 provinces base on a suite of climatological features. An optimal parameter set from a set of 33 predictors is determined for each province. The authors use this knowledge to create a monthly product of ocean pCO2 from 1992-2019 at a 1x1 spatial resolution. Identifying optimal parameters is useful, especially for high-resolution regional products. Using a NN-based stepwise regression technique to identify the parameters is novel and something I have not seen before. I think this manuscript is a useful contribution to the field. However, the manuscript needs some improvements. The manuscript is well organized, but moving some text to tables and rearranging some paragraphs will make the manuscript easier to follow. The figures are appropriate but the figure legends need more clarifying text.

**Below are specific line comments.**

L38 : What are the differences and how were the estimates made?

L41 : I would consider rephrasing the "Surface ocean pCO2 is …" sentence to something like "The magnitude and direction of the flux is largely set by the air-sea pCO2 difference." I think this is a nice lead-in to the next sentence. I would avoid saying "in the data-based method" because this is something that is true in the real world too.

L64-66: Consider expanding on this idea and explaining why each feature was chosen. Each feature can be considered a proxy for a process influencing pCO2:

SST and SSS --> solubility
Chl-a --> phytoplankton uptake
MLD --> entrainment
xCO2 --> Henry's law

I think a description of this will be useful for some readers

L66-78 : A table could make this list of features easier to read. I suggest a table of the features, references that use each feature, and maybe the physical process that each feature is a proxy for.

L100: I think "conversion" is more appropriate than "transition" here.

L102 : I like that you included units for the gas constant, please include units for each term (pCO2, fCO2, P, etc.)

L106: I am unsure what "parts of indicators" means. I think this can be removed and replaced with something like "Predictors used in this study were chosen from previously published ocean pCO2 products."

L109: Should this be Cheng et al. (2017)?
https://www.science.org/doi/10.1126/sciadv.1601545

L109-122 : consider putting these features into a table for ease of reading.

L119: This is just a note that ERA interim has been deprecated in favor of ERA5.

L135 : Why were 12 provinces chosen?

L138: Please be specific here. How were island provinces defined? Having less than X pixels? For completeness, please indicate where this island province was and what it was merged with. How were island provinces quantified? Having less than X pixels? Maybe a better phrasing is something like: "SOM-based provinces needed to meet the following criteria: 1. contain more than X pixels. 2. co-locate with at least X SOCAT observations. Provinces that do not meet the criteria were merged with the dominant neighboring province.

L139: "provinces covering area separated by land.." please explain this or give an example.

L141: Is 200m a typical definition for the coast? Can you please point to other studies that use this definition or indicate why this was chosen.

L144: Have you tried different predictions to test this idea?

L145: Please clarify this sentence. I am unsure what this means.

L151: Consider replacing this with a definition of what the stepwise part means. I am not too familiar with stepwise regression and a couple of sentences describing what the stepwise part means could be beneficial to readers. Since this approach is integral to the paper it is important that it is defined well.

L200: This paragraph may be more appropriate at the beginning of this section

L210: does the result change significantly for depending on your choice of random number?

L225: could cite figure 4a. I am curious if you tried deeper networks with more than 1 layer?

L233: This is nit-picky, but I always get confused if "to 2019" means the product runs through 2019 or ends in December 2018. I would consider either changing to "through 2019" or being specific and putting months in as well.

L237: This is great, I am glad the approach is gaining momentum. Could cite Gregor et al. (2019), that is the first place I have seen individual years used to improve independence.

L253: Note that these datasets are not included in the SOCAT dataset since pco2 is estimated and not directly measured. It is important to note that this data is completely independent from SOCAT.

L297: consider changing "proved" to "provides evidence for". I am not surprised SST and SSS are important since the solubility is a large driver of pCO2.

L346: Make it clear this value is from your product

L355: remove obviously

L434: Maybe "have similar spatial patterns with high pCO2 in the eastern equatorial Pacific" is a better way to phrase this.

L474: I could not download the script or dataset. Please make sure these are available everywhere. Zenodo is a public repository to consider.

**Typos**

There may be more that I missed. Please read the manuscript carefully.

L41 : Surface

L60 : methods

L99 : pCO2 and predictors

L175: store

L178: calculate

**Figures:**

All the figures need more descriptive legends.

Fig. 1: This figure is very detailed. However, it's hard to identify where to start reading from and the legend is not detailed enough. For instance, the reader doesn't even know the difference between indicator pool and input pool from the figure alone and it is unclear what Endcheck and Eo represent. Consider either adding color to the diagram to make it easier to read or simplifying it.

Fig. 2: this is nice, a classic neural network diagram. However, add more details in the legend. To make it clear you could also add the equation below hidden layer and summation layer.

Fig. 3: Consider naming the provinces something meaningful instead of numbers. For instance, East Equatorial Pacific, North Pacific Subpolar, North Pacific Subtropical, etc. I found myself constantly referring back to this image and names like this will make the paper easier to follow.

Also, this looks similar to the Fay and McKinley biomes (https://essd.copernicus.org/articles/6/273/2014/essd-6-273-2014.html). I don't think this is necessary here, but I wonder if using 17 biomes could recreate the biomes?

Fig. 4: this is fine, just add more description. Figure (a) could even be moved to supplementary.

Fig. 5: Consider making the text larger on the colorbars. It is difficult to read.

Fig. 6: Consider moving this to supplementary. This figure doesn't add to the story.

Fig. 7: This is fine, the text could be larger, and consider removing the tick labels in the middle of the plot. I would also consider moving away from the rainbow colormap since it has abrupt color changes that are meaningless. Cmocean has nice colormaps and is available for python and matlab (https://matplotlib.org/cmocean/).

Fig. 8: This is fine.

Fig. 9: Consider replacing "previous climatology product" with "Landschützer et al. (2020) product" Also consider using a non-rainbow colormap. My suggestion is the thermal colormap in cmocean.

**Tables:**

Table 1,2 : these are nice, just more description.

Table 3: Consider changing the province names to something more descriptive so the reader doesn't have to constantly refer back to the figure.

Table 4: Make the lowest MAE and RMSE for each province stand out. Bold those values or shade the box. This will allow you to quickly see which FFNN performs best in each province

**References mentioned in this review**

Cheng L., K. Trenberth, J. Fasullo, T. Boyer, J. Abraham, J. Zhu, 2017: Improved estimates of ocean heat content from 1960 to 2015, Science Advances, 3, e1601545. https://advances.sciencemag.org/content/3/3/e1601545.

Fay, A. R., and G. A. McKinley. "Global open-ocean biomes: mean and temporal variability." Earth System Science Data 6.2 (2014): 273-284

Gregor, Luke, et al. "A comparative assessment of the uncertainties of global surface ocean CO 2 estimates using a machine-learning ensemble (CSIR-ML6 version 2019a)–have we hit the wall?." Geoscientific Model Development 12.12 (2019): 5113-5136.

---

## Author Comment (AC1)

**Reviewer 1**

The authors developed a new ML approach to reconstruct global surface ocean pCO2 that considers an impact of different predictors in different ocean regions. Based on Self-Organizing Map method authors defined 11 biogeochemical provinces. A stepwise FFNN regression algorithm was applied to each of these provinces to establish a set of predictors that are highly responsible for pCO2 variability in considered province. Based on selected predictors and analysis of FFNN size (number of neurons) a monthly 1°x1° surface ocean pCO2 product from 1992 to 2019 was constructed. The results show a good agreement with validation data and independent observations.

I found this work well-organized and easy to read. It was interesting to see new predictors (phosphate, nitrate, silicate, dissolved oxygen) and their role in pCO2 variability. The authors presented important results for the Indian Ocean where due to the lack of observations different methods show their disagreements.

*Response: Thank you very much for your appreciation and very valuable suggestions to improve the manuscript!*

Below, I listed several points that need to be clarified before publication. Comments:

■ Boundaries between provinces. In the text we can find "To obtain a smoother distribution, we defined that the grid within 5 1x1 grids of province borders belong to all provinces adjacent to the nearest province border. Samples in these grids were involved in the FFNN training process of multiple provinces, but only counted once in the validation." Please could you clarify what you mean by "only counted once in the validation"? Is only an output from one province used in the validation? If yes, how do you chose a province from which you take an output?

*Response: Thank you for pointing out this unclear description. Due to the definition of new boundaries, in each province additional samples were added, which was outside the original boundary, referred as 'boundary sample' here. Now each province contains two types of samples: original samples and boundary samples. The boundary samples were only involved in the training process and were not set as validation samples in the province that it was defines as boundary samples. For one sample near the boundary, it is a 'original sample' in only one province and is a 'boundary sample' in other provinces. Thus, the sample was involved in the validation of only one province, and was involved in the training process in other provinces as 'boundary sample'. The*

*text was modified as "To obtain a smoother distribution, we extended the boundaries of all provinces 5 1°×1° grids outside and divided the samples inside and outside the original boundary of each province into 'original sample' and 'boundary sample'. For one sample near the boundary, it is a 'original sample' in only one province and is a 'boundary sample' in other provinces. Thus, the sample was involved in the validation of only one province, and was involved in the training process in other provinces as 'boundary sample'."*

■ **Independent observations. Please could you provide geographical positions and period of stations used as independent observations?**

*Response: The HOT station is located in 22° 45'N, 158° 00'W and observations started since October 1988. The BATS station is located in 31°50'N, 64°10'W and observations are from October 1988 to December 2019. The ESTOC station is located in 29°10'N, 15°30'W and observations from 1995 to 2009 were used. The information above was added in the validation section.*

■ **Set of selected predictors. In table 3 authors presented two sets for most of the regions that depends on the availability CHL-a data. Please could you present more explicitly that the final product is built on two FFNNs, one trained for the period 1992-2001 based on one predictors set and another – for 2002-2019 based on the second predictors set?**

*Response: Thanks for your suggestion. A description was added in the section 2.4 pCO2 product as "Then the final product was built based on two FFNNs, one trained for the period 2002-2019 using one predictor set including CHL-a or CHL-a anom, and the second one for the 1992-2001 using the second predicator set without CHL-a and CHL-a anom."*

■ **More explicit figures' captions. Please provide more explicit figures' captions, period of presented results, or results averaged over xxxx-xxxx, what are horizontal lines in Fig.6b?**

*Response: Thanks for your suggestion. The horizontal line was the average pCO2 growth rate over each decade (1992-2000, 2001-2010 and 2011-2019).*

■ **Not correct conclusion. On page 15 lines 375-379 authors concluded that the difference between FNN1 and FFNN3 is relatively small, because predictors used in FFNN1 and FFNN3 were related to main drivers of pCO2, such as CHL-a, xCO2 and MLD. However, same drivers are used in FFNN2. Thus, it cannot explain why FFNN2 shows higher differences with observations.**

*Response: Thank you for pointing out this mistake. After reconsidering this issue, I think the application of latitude and longitude as predicators of pCO2 may be the reason why FFNN2 shows higher MAE and other validation groups shows relatively closer results. For example, in the province P10 that latitude and longitude were considered not good predictors by the stepwise FFNN algorithm, the three validation groups show significant closer results than that in other provinces. While in other provinces, latitude and longitude were used as predictors in the FFNN1 and FFNN3, decreasing the MAE and RMSE. The text was corrected as "The MAE and RMSE difference between FFNN1 and FFNN3 in some provinces were relatively small. The reason for higher MAE and RMSE showed by the FFNN2 may be the application of latitudes and longitudes as predicators in both the FFNN1 and FFNN3 but not in the FFNN2. In the province P10, latitudes and longitudes were considered not good predictors by the stepwise FFNN algorithm and the results of three validation groups were extremely close.".*

■ **On page 18 line 430 authors said that the pattern of reconstructed pCO2 climatology was close to SOCAT in the Indian Ocean. I would say that it is not so close to mention it in this sentence.**

*Response: Thanks for your suggestion. The inaccurate description was now removed.*

■ **Page 10 lines 298-300: For better structure of paragraph the sentence "In the province P1 located in the Arctic, the silicate concentration and temperature were considered as the most crucial predictor of pCO2." could be moved at the end of paragraph where authors mentioned the phosphate, nitrate, silicate, etc.**

*Response: Thanks for your suggestion. The sentence was now moved at the end.*

■ **Page 13 lines 336-337: The sentence about results in the Indian Ocean can be removed if you put the Indian Ocean in the previous sentence, or please add "Also" at the beginning of the sentence dedicated to results in the Indian Ocean.**

*Response: At the beginning of the sentence "Also" has been added now.*

■ **Page 16 lines 390-392: Please could you reformulate this sentence ("The interannual variability and seasonal pattern..") as it is difficult to read?**

*Response: The sentence was modified as "Compared with the independent*

*observations from the HOT station, the three validation groups both show close results, which were also similar with each other in the seasonal and interannual variability of pCO2".*

■ **Page 18 lines 432-436: two sentence can be combined: "Compared with previous climatology product (Landschuster et al., 2020), the global distribution pattern of surface ocean pCO2 was basically well consistent: inconsistent spatial distribution also existed in the Arctic and parts of the Southern Ocean near the Antarctic continent."**

*Response: Thanks for your suggestion. The two sentences now have been combined.*

**Typo:**
**Page 2 line 41: "surface ocean pCO2" should be replaced by "Surface ocean pCO2".**
**Page 9 line 261: "validation group" should be replaced by "Validation group".**
**Page 13 line 332: "Based the K-fold" should be replaced by "based on the K-fold".**
**Page 20 line 459: "based improved FFNN size" should be replaced by "based on improved FFNN size".**

*Response: Thank you for pointing out these mistakes. These mistakes have been corrected now.*

---

## Author Comment (AC2)

**Review 2**

**General comments:**
**The stepwise FFNN looks like a innovative new approach to enhance the widely popular SOM-FFNN method. The stepwise method is tested using a very comprehensive list of predictors used elsewhere in the literature. The method building component looks thorough, congrats. The prediction of pCO2 based on region-specific predicators selected by the stepwise FFNN algorithm will be a valuable tool when moving to higher resolution, inside regional studies, or getting closer to shore. There are a number of grammatical errors that will need to be cleaned up by the authors or the journal team.**

*Response: Thank you very much for your appreciation and very valuable suggestions to improve the manuscript!*

**Specific comments:**
**1 Introduction**
**Line 66-82: Appreciate the list of previous works and predictor data used by each, provides justification for use in the stepwise FFNN. Would like to see one more sentence relating use of different predictors leading to varying marine sink estimates.**

*Response: Thanks for you suggestion. In previous researches, not only different predictors, but also different methods were used. Thus, the differences in estimate of marine carbon sink between previous researches were not only caused by use of different predictors. In addition, there is almost no such research that focusing on the influence of predictor differences on marine sink estimate.*

**2 Methodology**
**2.1 Data**
**Line 106-122: Are all these products retrieved at the same resolution? Are they upscaled or downscaled at all to your needs?**

*Response: Most of these products were retrieved at 1° × 1° resolution. Some products retrieved at higher resolution were downscaled to 1° × 1° resolution. This description has been added at the end of the 2.1 Data section.*

**Line 122: "In addition, 8 parameters…." Thanks for listing after this sentence. Which previous research used these as predictors in observation-based pCO2 estimates? List and provide citations like in the introduction. Or is the inclusion of these predictors' novel? If so, highlight that.**

*Response: These listing 8 parameters have not been used as predictors in observation-based pCO2 estimates in previous researches yet, but nutrients and dissolved oxygen have been used as predictors in observation-based estimates of total alkalinity and DIC. The citation has been added.*

**2.2 Biogeochemical provinces defined by the Self-Organizing Map**
**Line 134: These SOM predictors exclude most of the FFNN predictors discussed in the introduction. Was there a reason why "biological" predictors (i.e., nutrients and oxygen) are weighted so heavily in the SOM selection? Were more physical predictors (i.e., mixed layer depth, etc.) used in SOM testing to optimize provinces? Or similar to previous work (Landschützer), using published pCO2 climatology as a predictor to determine provinces?**

*Response: Thanks for noting this problem. Actually, we also used mixed layer depth, sea surface height and pCO2 climatology from Landschützer, 2020, but mistakenly lost in the text. The description about predictors has been corrected now.*

**Line 135: Just out of curiosity, did the configuration (3-by-4 size) make much of a difference to SOM province distribution?**

*Response: In the early work, 4-by-4 or 4-by-5 size were also attempted. Increasing size led to appearance of small provinces inside main provinces, but the distributions of main provinces were similar. To simplify the SOM boundary issues, we choose the 3-by-4 size with less provinces.*

**Line 141: The 200 m depth boundary is fairly close to shore. Is this a commonly used open oceanic / coastal ocean boundary? If so citations from other studies here.**

*Response: It is not a commonly used boundary. In previous researches focusing on coastal pCO2 reconstruction, the boundary was defined as 1000m depth or 300 km offshore. We defined the boundary as 200m depth because the SOCAT samples with high predicting error were mainly located in areas shallower than 200m.*

**Line 144: Unique way to address the SOM boundary issue. Cool.**

*Response: Thank you for your appreciation.*

**2.3 Stepwise FFNN algorithm**
**Line 152-163: Clarify. Was the mean absolute error used for the internal**

**MATLAB neural network performance loss function (in the training targets and validation targets steps used to end training), also / or as a means for evaluating the FFNN output pCO2 product to withheld data?**

*Response: The MAE was used for performance loss function, and also in the validation of pCO2 product using a K-fold cross validation method.*

**Line 172: "…referred to as indicators pool (Start in Fig. 1),…" Keep coming back to this Figure throughout if you can. Makes it easier to read and connect to the Figure.**

*Response: Thank you for your suggestion. More annotations have been added in the description.*

**Figure 1: More sub boxes (dotted lines), connected to text could also make it easier to follow. Steps between loop 1 and loop 2, steps between loop2 and end.**

*Response: Thank you for your suggestion. More sub boxes have been added in the Figure 1.*

**2.4 pCO2 product**
**Line 227: Reason why 10 and 70 are chosen? Is there a possibility that even in smaller provinces 10 neurons could lead to overfitting? The polar regions are set right at 10.**

*Response: We test the number of neurons from 5 to 300. The MAE continuously increasing after 70. The variation of MAE would be difficult to see clearly if all spots were showed, so the results after 70 were omitted. Seems the way the result shows may be misleading, so we redrew the Fig.4a. Not just focusing on overfitting, too few neurons may lead to insufficient learning capacity for complex nonlinear relationship, so we tested the performance of FFNN with different number of neurons.*

**Line 231: Does this vary neuron number test really limit overfitting? Taking the lowest MAE from the internal train/validation split during FFNN training step just means it is likely replicating training data well and due to random split inside autocorrelated validation data this doesn't change much. Being clear in Line 152-163 about how / when MAE is used to evaluate could clear this up.**

*Response: The MAE used here was calculated using a K-fold cross validation method grouping by year. The training data and validation data were taken from different years and were relatively independent. The MAE theoretically tend to*

*increase when insufficient learning capacity due to too few neurons or overfitting problem due to too many neurons appear. The result shows that the MAE did increase when the number of neurons was lower than 10 and higher than 100.*

**2.5 Validation**
**Line 237: Unique use of the K-fold cross validation method grouping by year.**

*Response: Since samples within 500 km in the same period were correlated, grouping by year makes the training data and validation data relatively more independent.*

**3 Results and discussion**
**3.1 Biogeochemical provinces and corresponding predictors of pCO2**
**Table 3: Add more to the caption on the order of the predictors listed.**

*Response: Thank you for your suggestion. The caption has been modified.*

**3.2 pCO2 product**
**Line 317: Does this mean you first went through the stepwise FFNN process using the same neuron number? Then when the best predictors where determined you used the varying neuron number test (from 10-70) to find the best neuron number? Then you used the K cross validation to test robustness? Clarify. Link to back to Figure 1 if you need to.**

*Response: Yes, the stepwise FFNN process use the same neuron number. Since the result of varying neuron number test shows there is almost no insufficient learning capacity or overfitting problem when the number of neurons was in 10-70 and the MAE differs a little. Any number of neurons in this range was considered suitable. Although a loop of "stepwise FFNN – neuron number test – stepwise FFNN - …." to use different number of neurons in the stepwise FFNN process may further decrease the predicting error, the effect was not so significant and a stable end is not easy to find. In the future work the role of the varying neuron number test may be reconsidered, but now it is used for avoiding insufficient learning capacity or overfitting problem in spite of the low possibility of appearance, and decreasing the predicting error slightly.*

**Figure 4: Still not sure on this test limiting overfitting. Looks like they all (except at the poles maybe because it is not well constrained…? As in your Table 4) just level out. Using the same FFNN predictors and the same targets how reproducible is this Figure? Or is it dependant on the initialization on that run?**

*Response: The Fig. 4a has been modified to show the additional result of 70-300. The MAE increased after 100. In MATLAB we used "setdemorandstream(pi)" to set initial state stable, thus the result using the same FFNN predictors and the same targets is completely reproducible.*

[Figure]

**Line 334: Good to state this up front. Other than these regions it does look good. However, if the goal from the introduction is get at the air-sea flux, how important are these regions for the global marine CO2 flux? Suggest in conclusions what could be done in the future to improve these regions?**

*Response: The east equatorial Pacific is the most important CO2 source, while the subpolar Pacific was a sink in summer and a source in winter. The CO2 flux in the Southern Ocean near the Antarctic continent was near zero due to ice cover. For the future work to improve these regions, maybe more parameters related to biological activities, El Nino and La Nina, or remote sensing parameters will be added to constrain the pCO2 in these regions.*

**Figure 6a: Would be nice to also have the atmospheric xCO2 product on this Figure for comparison.**

*Response: Thank you for your suggestion, the atmospheric CO2 has been added in the Fig. 6a.*

**3.4 Validation based on independent observations**
**Line 395: Nothing is obvious to every reader. Remove and clarify.**

*Response: Thank you for your suggestion. The unbefitting description has been removed.*

**Line 442: "… was credible." Is consistent with and improves upon? Readers should want to believe in what you did! Got to sell it a bit!**

*Response: Thank you for your suggestion. The text has been modified as "suggesting that pCO2 predicting based on regional different predictors*

*selected by the stepwise FFNN algorithm was better than that based on the globally same predictors."*

**4. Conclusions**
**Line 447-465: This needs a bit of a rework. Feels like recycled sentences from throughout. What should readers take away from your work? How can this approach be applied in other studies? Who benefits from this improvement? Where is more work needed (e.g., polar regions), how could improvements be made?**

*Response: Thank you for the suggestion. This part has been rewritten as "A stepwise FFNN algorithm was constructed to decreasing the predicating error in the surface ocean $pCO_2$ mapping by finding better combinations of $pCO_2$ predicators in each biogeochemical province defined by SOM method, based on which a monthly 1°×1° gridded global open-oceanic surface ocean $pCO_2$ product from January 1992 to August 2019 was constructed. Our work provided a statistical way of predictor selection for all researches based on relationship fitting by machine learning methods, and shows that using regional-specific predictors selected by the stepwise FFNN algorithm retrieved lower predicting error than using globally same predictors. This stepwise FFNN algorithm can be also used in $pCO_2$ mapping researches for higher resolution and coastal regions, and also in other data mapping researches using SOM or other region dividing method. The prepare work was only collecting as many parameters, which are possibly related to the target data and need to be sufficiently available in time and space. However, high predicting error in special regions still remains to be improved, such as polar regions and equatorial Pacific. Since the result of the stepwise FFNN largely depends on the way biogeochemical provinces divided, improving of SOM step is still necessary. Besides, the FFNN can be replaced by any suitable type of neural networks. A possible way to improve the performance of stepwise FFNN algorithm is to modify the structure of FFNN or to use better networks.".*

---

## Author Comment (AC3)

**Reviewer 3**

This manuscript uses a stepwise feed-forward neural network (FFNN) to identify an optimal feature for the prediction of ocean pCO2. The authors first use a self-organizing map (SOM) to cluster the ocean into 12 provinces base on a suite of climatological features. An optimal parameter set from a set of 33 predictors is determined for each province. The authors use this knowledge to create a monthly product of ocean pCO2 from 1992-2019 at a 1x1 spatial resolution. Identifying optimal parameters is useful, especially for high-resolution regional products. Using a NN-based stepwise regression technique to identify the parameters is novel and something I have not seen before. I think this manuscript is a useful contribution to the field. However, the manuscript needs some improvements. The manuscript is well organized, but moving some text to tables and rearranging some paragraphs will make the manuscript easier to follow. The figures are appropriate but the figure legends need more clarifying text.

*Response: Thank you very much for your appreciation and very valuable suggestions to improve the manuscript!*

**Below are specific line comments.**

**L38: What are the differences and how were the estimates made?**

*Response: The average global ocean sea-air $CO_2$ flux estimated by sea-air $pCO_2$ differences using different $pCO_2$ products differ from -1.55 to -1.74 PgC $yr^1$ during 2001-2015, and the differences in individual years reached nearly 0.6 PgC $yr^{-1}$(Rödenbeck et al., 2014; Iida et al., 2015; Landschützer et al., 2014; Denvil-Sommer et al., 2019). These estimates were made by multiplying sea-air $pCO_2$ differences by piston velocity, seawater density and $CO_2$ solubility, based on $pCO_2$ products constructed using statistical interpolation or machine learning methods. More specific description was added in the manuscript.*

**L41: I would consider rephrasing the "Surface ocean pCO2 is …" sentence to something like "The magnitude and direction of the flux is largely set by the air-sea pCO2 difference." I think this is a nice lead-in to the next sentence. I would avoid saying "in the data-based method" because this is something that is true in the real world too.**

*Response: The sentence has been rephrased according to the suggestion.*

**L64-66: Consider expanding on this idea and explaining why each feature was chosen. Each feature can be considered a proxy for a process**

influencing pCO2:
SST and SSS --> solubility
Chl-a --> phytoplankton uptake
MLD --> entrainment
xCO2 --> Henry's law
I think a description of this will be useful for some readers

*Response: Thank you for the suggestion. Additional description about the selection of each feature has been added.*

**L66-78 : A table could make this list of features easier to read. I suggest a table of the features, references that use each feature, and maybe the physical process that each feature is a proxy for.**

*Response: A table has been added in the supplementary, listing all features used and describing the references using the feature, data products used, spatial and temporal coverage.*

**L100: I think "conversion" is more appropriate than "transition" here.**

*Response: Thank you for the suggestion. The word "transition" was replaced.*

**L102: I like that you included units for the gas constant, please include units for each term (pCO2, fCO2, P, etc.)**

*Response: The units were added in the description. The sentences were modified as "where $fCO_2$ and $pCO_2$ are in micro-atmospheres (µatm), P is the total atmospheric surface pressure (Pa) using the National Centers for Environmental Prediction (NCEP) monthly mean sea level pressure product (Dee et al., 2011), and T is the absolute temperature (K). R is the gas constant (8.314 J $K^{-1}$ $mol^{-1}$). Parameters B ($m^3$ $mol^{-1}$) and δ ($m^3$ $mol^{-1}$) are both viral coefficients (Weiss, 1974)."*

**L106: I am unsure what "parts of indicators" means. I think this can be removed and replaced with something like "Predictors used in this study were chosen from previously published ocean pCO2 products."**

*Response: This selection was supposed to show most predictors used in this work were chosen from previously published ocean $pCO_2$ products, and some predictors were first used in the $pCO_2$ reconstructing. The sentence has been modified as "In this work, total 33 indicators were used. Where 25 indicators were chosen from previous researches of surface ocean pCO2 reconstruction …"*

**L109: Should this be Cheng et al. (2017)? https://www.science.org/doi/10.1126/sciadv.1601545**

*Response: The citation of temperature data is Cheng et al. (2016) and Cheng et al. (2017), and the citation of salinity data is Cheng et al. (2020). The citation has been corrected.*

**L109-122: consider putting these features into a table for ease of reading.**

*Response: Thank you for the suggestion. A table has been added in the supplementary, listing all features used and describing the references using the feature, data products used, spatial and temporal coverage.*

**L119: This is just a note that ERA interim has been deprecated in favor of ERA5.**

*Response: Because the temporal coverage of pCO2 product in current version was only in 1992-2019. The ERA5 product will be used instead of ERA interim in the future version when other data product is sufficiently available for the reconstruction of pCO2 after 2019.*

**L135: Why were 12 provinces chosen?**

*Response: In the early work, different number of provinces such as 16 or 20 were also attempted. Increasing number led to appearance of small provinces inside main provinces, but the distributions of main provinces were similar, such as provinces covering north Pacific, north Atlantic, equatorial and polar areas. In addition, more provinces lead to less SOCAT samples in each one province. So, we used as few as possible provinces to make sure that there are sufficient training samples in each one province.*

**L138: Please be specific here. How were island provinces defined? Having less than X pixels? For completeness, please indicate where this island province was and what it was merged with. How were island provinces quantified? Having less than X pixels? Maybe a better phrasing is something like: "SOM-based provinces needed to meet the following criteria: 1. contain more than X pixels. 2. co-locate with at least X SOCAT observations. Provinces that do not meet the criteria were merged with the dominant neighboring province.**

*Response: Thank you for the suggestion. Provinces with connected pixels less than 10 and provinces with SOCAT observation less than 1000 were define as island provinces, and then merged with nearest provinces. The more specific*

*description has been added.*

**L139: "provinces covering area separated by land." please explain this or give an example.**

*Response: The province P3 covering north temperate Pacific and the province P5 covering north temperate Atlantic were set as one province in the original output of SOM, but were mainly separated by The North American continent. So, we divided the province into two new provinces. Same process was carried out in the northwest Pacific, Mediterranean and so on. The more specific description has been added.*

**L141: Is 200m a typical definition for the coast? Can you please point to other studies that use this definition or indicate why this was chosen.**

*Response: It is not a widely used definition and different definition were used in previous researches. For example, 1000m depth and 30 salinity as boundary was used in Zeng et al., 2014, and 500m depth as boundary was used in Telszewski et al., 2009. Researches focusing on coastal pCO2 used a boundary of 1000m depth/300km offshore (Laruelle et al., 2017). We used 200m depth as boundary because the grids with high predicting error were mainly located in areas <200m depth.*

**L144: Have you tried different predictions to test this idea?**

*Response: We have compared the result using different predicators with the result using same predicators in all provinces. more obvious border lines appeared in some regions when using different predicators in each province, but we are not sure whether it is caused by application of a certain predicator or by the differences of predictors between neighboring provinces.*

**L145: Please clarify this sentence. I am unsure what this means.**

*Response: We extended the boundaries of all provinces 5 1°×1° grids width outside. In each one province, samples near the province boundary but belong to other province were also involved in the training process. For example, if province P1 and P2 are neighboring, samples belong to province P2 near the boundary of P1 were also used in the training of FFNN of P1. The distribution of pCO2 became smoother after this definition of province boundary was used.*

**L151: Consider replacing this with a definition of what the stepwise part means. I am not too familiar with stepwise regression and a couple of sentences describing what the stepwise part means could be beneficial to readers. Since this approach is integral to the paper it is important that**

**it is defined well.**

*Response: Thank you for the suggestion. The sentence was replaced by "In the stepwise part, predicators of $pCO_2$ are going to be added and removed one by one, and which predicators will be finally used in the $pCO_2$ predicting is determined according to the real-time change of predicating error."*

**L200: This paragraph may be more appropriate at the beginning of this section**

*Response: Thank you for the suggestion. The paragraph has been moved to the beginning.*

**L210: does the result change significantly for depending on your choice of random number?**

*Response: The way that initial bias and weights matrixes of a FFNN randomly assigned depends on the random number stream. The result basically changed slightly when the initial state or the way testing sample group divided changed. For example, if 10 predictors were selected in the stepwise part, the last 2-3 predictors may change when the initial state of FFNN changed.*

**L225: could cite figure 4a. I am curious if you tried deeper networks with more than 1 layer?**

*Response: We test FFNN with two hidden layers. The result when using two hidden layers and 25 neurons in each layer was similar with the result using 125 or more neurons in one hidden layer. But we did not test more neurons in two hidden layer or more hidden layers, because testing of one province takes over one week or even longer.*

**L233: This is nit-picky, but I always get confused if "to 2019" means the product runs through 2019 or ends in December 2018. I would consider either changing to "through 2019" or being specific and putting months in as well.**

*Response: Thank you for the suggestion. The specific months was added.*

**L237: This is great, I am glad the approach is gaining momentum. Could cite Gregor et al. (2019), that is the first place I have seen individual years used to improve independence.**

*Response: The citation has been added.*

**L253: Note that these datasets are not included in the SOCAT dataset since pco2 is estimated and not directly measured. It is important to note that this data is completely independent from SOCAT.**

*Response: Thank you for the suggestion. More description was added.*

**L297: consider changing "proved" to "provides evidence for". I am not surprised SST and SSS are important since the solubility is a large driver of pCO2.**

*Response: The unproper description has been changed to "provides evidence for".*

**L346: Make it clear this value is from your product**

*Response: Thank you for the suggestion. The description was modified as "The global open ocean average $pCO_2$ of the product generated in this work increased about 1.85 µatm per year".*

**L355: remove obviously**

*Response: The unproper description has been removed.*

**L434: Maybe "have similar spatial patterns with high pCO2 in the eastern equatorial Pacific" is a better way to phrase this.**

*Response: Thank you for the suggestion. The description was modified.*

**L474: I could not download the script or dataset. Please make sure these are available everywhere. Zenodo is a public repository to consider.**

*Response: The website was supposed to be globally available. I am not sure if the full stop of the last sentence was misleading. The website is http://dx.doi.org/10.12157/iocas.2021.0022 without a dot at the end. If the download page is still not available in your region, we will use Zenodo as a second repository, because this work and the MSDC repository belongs to a same research program and the product is planned to be stored at the MSDC repository.*

**Typos**
**There may be more that I missed. Please read the manuscript carefully.**
**L41 : Surface**
**L60 : methods**

L99 : pCO2 and predictors
L175: store
L178: calculate

*Response: Thank you for pointing out the typos. We noticed that in the manuscript the word "predictor" and "predicator" were totally confused. Now the typos were corrected.*

**Figures:**
**All the figures need more descriptive legends.**
**Fig. 1: This figure is very detailed. However, it's hard to identify where to start reading from and the legend is not detailed enough. For instance, the reader doesn't even know the difference between indicator pool and input pool from the figure alone and it is unclear what Endcheck and Eo represent. Consider either adding color to the diagram to make it easier to read or simplifying it.**

*Response: Thank you for the suggestion. More legends and descriptions were added in the figure.*

**Fig. 2: this is nice, a classic neural network diagram. However, add more details in the legend. To make it clear you could also add the equation below hidden layer and summation layer.**

*Response: The equation has been added in the figure.*

**Fig. 3: Consider naming the provinces something meaningful instead of numbers. For instance, East Equatorial Pacific, North Pacific Subpolar, North Pacific Subtropical, etc. I found myself constantly referring back to this image and names like this will make the paper easier to follow. Also, this looks similar to the Fay and McKinley biomes (https://essd.copernicus.org/articles/6/273/2014/essd-6-273-2014.html). I don't think this is necessary here, but I wonder if using 17 biomes could recreate the biomes?**

*Response: Thank you for the suggestion. The provinces name was changed to numbers following by locations. The Fay and Mckinley biomes used SST, CHL-a and MLD, which are also used in this work. If using 17 biomes maybe the result will be more similar. But we want to use a simpler province set to make sure that there are as many SOCAT samples in each province, because the result of stepwise FFNN was largely influenced by the input SOCAT samples.*

**Fig. 4: this is fine, just add more description. Figure (a) could even be moved to supplementary.**

*Response: More description was added.*

**Fig. 5: Consider making the text larger on the colorbars. It is difficult to read.**

*Response: The figure was redrawn to make the colorbars larger.*

**Fig. 6: Consider moving this to supplementary. This figure doesn't add to the story.**

*Response: Thank you for the suggestion. The figure has been moved to supplementary.*

**Fig. 7: This is fine, the text could be larger, and consider removing the tick labels in the middle of the plot. I would also consider moving away from the rainbow colormap since it has abrupt color changes that are meaningless. Cmocean has nice colormaps and is available for python and matlab ([https://matplotlib.org/cmocean/](https://matplotlib.org/cmocean/)).**

*Response: Thank you for the suggestion. The size of text was adjusted and the "balance" colormap from the Cmocean was used.*

**Fig. 8: This is fine.**
**Fig. 9: Consider replacing "previous climatology product" with "Landschützer et al. (2020) product" Also consider using a non-rainbow colormap. My suggestion is the thermal colormap in cmocean.**

*Response: Thank you for the suggestion. The title has been replaced. The thermal colormap in cmocean was used.*

**Tables:**
**Table 1,2: these are nice, just more description.**

*Response: More description was added.*

**Table 3: Consider changing the province names to something more descriptive so the reader doesn't have to constantly refer back to the figure.**

*Response: Thank you for the suggestion. The province names were changed to description of spatial locations.*

**Table 4: Make the lowest MAE and RMSE for each province stand out.**

**Bold those values or shade the box. This will allow you to quickly see which FFNN performs best in each province**

*Response: Thank you for the suggestion. The values were highlighted in bold.*

**References mentioned in this review**
Cheng L., K. Trenberth, J. Fasullo, T. Boyer, J. Abraham, J. Zhu, 2017: Improved estimates of ocean heat content from 1960 to 2015, Science Advances, 3, e1601545. https://advances.sciencemag.org/content/3/3/e1601545.
Denvil-Sommer, A., Gehlen, M., Vrac, M., and Mejia, C.: LSCE-FFNN-v1: a two-step neural network model for the reconstruction of surface ocean pCO2 over the global ocean, Geoscientific Model Development, 12, 2091-2105, 10.5194/gmd-12-2091-2019, 2019.
Fay, A. R., and G. A. McKinley. "Global open-ocean biomes: mean and temporal variability." Earth System Science Data 6.2 (2014): 273-284
Gregor, Luke, et al. "A comparative assessment of the uncertainties of global surface ocean CO 2 estimates using a machine-learning ensemble (CSIR-ML6 version 2019a)–have we hit the wall?." Geoscientific Model Development 12.12 (2019): 5113-5136.
Iida, Y., Kojima, A., Takatani, Y., Nakano, T., Midorikawa, T., and Ishii, M.: Trends in pCO2 and sea-air CO2 flux over the global open oceans for the last two decades, J. Oceanogr., 71, 637–661, https://doi.org/10.1007/s10872-015-0306-4, 2015.
Landschutzer, P., Gruber, N., Bakker, D. C. E., and Schuster, U.: Recent variability of the global ocean carbon sink, Global Biogeochem. Cy., 28, 927–949, https://doi.org/10.1002/2014GB004853, 2014.
Rödenbeck, C., Bakker, D. C. E., Metzl, N., Olsen, A., Sabine, C., Cassar, N., Reum, F., Keeling, R. F., and Heimann, M.: Interannual sea–air CO2 flux variability from an observationdriven ocean mixed-layer scheme, Biogeosciences, 11, 4599– 4613, https://doi.org/10.5194/bg-11-4599-2014, 2014.

---

## Referee Report (RR1)

The manuscript is refined compared to the original version and all of my comments were properly addressed. This work is a useful contribution to the field because this is the first time I have seen the regional importance of predictors addressed in a rigorous way.

The text could still use some minor improvements, mainly in the introduction. I have provided some detailed suggestions below. Once the text has been modified, then I think the manuscript is ready for publication

**Suggestions:**
Line 36: consider removing "have been thought to".

Line 38-44: Please rewrite this sentence for clarity. Maybe something like this: "However, the air-sea CO2 flux averaged between 2001-2015 varies from -1.55 and -1.74 PgC/yr, depending on the pCO2 product. These differences largely stem from differences in pCO2 estimates across the products."

Line 45-47: Consider changing sentence to: "Surface water pCO2 greater than the overlying air indicates CO2 is released from the ocean to the air. Conversely, absorption of CO2 by oceans happened when the pCO2 of the surface water is lower than the overlying air"

Line 48: need a comma after "sink".

Line 48: sentence starting with "Sparse and uneven…" could be a new paragraph.

Line 56: Consider changing "Recent researches on" to "Advances in"

Line 62: consider removing "methods such as" and changing "and other" to "with"

Line 68 " were considered as" to "are considered"

Line 74: "appeared" to "appears"

Line 77: Consider writing sentence as "In addition, sampling information, such as latitude and longitude, has been used as a predictor." Also, consider referencing Gregor et al. 2019

Lines 111-137: I would consider removing this text and moving Table S1 to the main text. Tables are much easier to read.

Line 145: Landschützer et al. (2020)

Line 148: consider rephrasing "Provinces with connected pixels less than 10 and provinces with SOCAT observation less than 1000…" to "Provinces with less than 10 pixels and less than 1000 SOCAT observations…"

Line 153: "The" needs to be lower case

Line 156: "200 m"

Line 167: comma after "summation".

Line 170: "a tan-sigmoid"

Line 173: Maybe remove "pure" from "pure linear function", I am not sure what that means.

Line 177: Please state how sensitive the model is to the choice of random number. That random number defines where the NN starts "searching" in errors space. I understand the random number was fixed for reproducibility purposes. However, my understanding of best practice is to either run the model many times with different random numbers and take the average or "tune" the random number to find the one that gives best results. Please mention if this was explored or if the random number was simply chosen randomly.

Line 224: "after that,"

Line 226: "Then,"

Line 231-235: Please consider rephrasing, this sentence is hard to understand.

Line 236: "At this time, part 1 … finished and …"

Line 259: In what increments where neurons increased by?

Line 290: It is also worth noting pCO2 at BATS and HOT are estimated from TA and DIC (I am not sure if ESTOC is estimated or directly measured). I think it's important for the reader to know these estimates are derived and not directly measured like SOCAT observations.

Line 505: Wouldn't modifying the structure of the FFNN be considered making a "better network"? By better network do you mean more sophisticated architectures? I would also add that different learning algorithms could be considered.

Table 4: Please state what the bold values indicate. Also a comma is needed before respectively: "…Landschützer et al., 2014 and Denvil-Sommer et al., 2019, respectively" This typo is repeated throughout the article

Figure 3: Make it clear these provinces are from a SOM. This is just a suggestion, but it would be great to get some of the information in Table 4 into this map. Maybe putting the leading predictor in parentheses in the colorbar? This might look too messy, it's just a suggestion.

Figure 6: Comma before "respectively"

Figure 7: Comma needed: "…Landschützer et al., 2014 and Denvil-Sommer et al., 2019, respectively

Figure 8: please put units on the colorbar.

---

## Referee Report (RR2)

**Comments on revised version.**

I would like to thank the authors for the changes they made. I am appreciated that the authors took into account my previous comments. The manuscript is more clear and completed now.

Comments:

Lines 38-44: New sentence is too long with repetitions, please modify it: "However, due to large uncertainty in estimates of surface ocean partial pressure of CO2 (pCO2), the long-term average global ocean sea-air CO2 flux during 2001-2015 estimated based on sea-air pCO2 difference differ from -1.55 to -1.74 PgC yr$^{-1}$, and the maximum difference between global sea-air CO2 flux in individual years reached nearly 0.6 PgC yr$^{-1}$ (Rödenbeck et al., 2014; Iida et al., 2015; Landschützer et al., 2014; Denvil-Sommer et al., 2019)". It can be something like: However, due to large uncertainty in estimates of surface ocean partial pressure of CO2 (pCO2), the long-term average global ocean sea-air CO2 flux during 2001-2015 differs from -1.55 to -1.74 PgC yr$^{-1}$ with the maximum difference in individual years nearly 0.6 PgC yr$^{-1}$ (Rödenbeck et al., 2014; Iida et al., 2015; Landschützer et al., 2014; Denvil-Sommer et al., 2019)."

Line 123: Need a space between "Interim" and "(Dee et al…"

Line 128: You use here a word "parameters" to call "predictors" or "indicators". It appears further in the text as well (for example, in Conclusion, line 498). I think it can be misunderstood as parameters can be used to explain FFNN architecture. Please keep using words "predictors" or "indicators" throughout the text.

Line 138: Please indicate which method you used for downscaling.

Lines 184-187: Please change this sentence, it is difficult to read: "The mean absolute error (MAE) difference that before and after adding or removing one indicator in the input of FFNN calculated using a K-fold cross validation method was used to estimate the performance of each indicator in the FFNN predicating."

Lines 318-320: Please combine these two sentences, it sounds like a repetition: "For example, month was considered as a recommended predictor in most provinces. Especially in the province P4 subpolar Atlantic and P5 north subtropical Atlantic, the parameter month was relatively more recommended."

Line 322: Please add "the sine of latitude".

Line 404: Remove point between "Table" and "4".

---

## Author Response (AR2)

**Response to reviewers – second review**

**Comments by reviewer Anna Denvil-Sommer**

The manuscript is refined compared to the original version and all of my comments were properly addressed. This work is a useful contribution to the field because this is the first time I have seen the regional importance of predictors addressed in a rigorous way.

The text could still use some minor improvements, mainly in the introduction. I have provided some detailed suggestions below. Once the text has been modified, then I think the manuscript is ready for publication

Suggestions:

■ Line 36: consider removing "have been thought to".
**Response**: *The words was now removed.*

■ Line 38-44: Please rewrite this sentence for clarity. Maybe something like this: "However, the air-sea $CO_2$ flux averaged between 2001-2015 varies from -1.55 and -1.74 PgC/yr, depending on the $pCO_2$ product. These differences largely stem from differences in $pCO_2$ estimates across the products."
**Response**: *Thank you for the suggestion. The sentence was modified as "However, the global ocean sea-air $CO_2$ flux averaged between 2001-2015 varies from -1.55 to -1.74 PgC yr$^{-1}$ with the maximum difference in individual years nearly 0.6 PgC yr$^{-1}$, depending on the surface ocean partial pressure of $CO_2$ ($pCO_2$) product. These differences largely stem from differences in $pCO_2$ estimates across the products."*

■ Line 45-47: Consider changing sentence to: "Surface water $pCO_2$ greater than the overlying air indicates $CO_2$ is released from the ocean to the air. Conversely, absorption of $CO_2$ by oceans happened when the $pCO_2$ of the surface water is lower than the overlying air"
**Response**: *Thank you for the suggestion. The sentence was changed as the suggestion.*

■ Line 48: need a comma after "sink".
**Response**: *The comma was added.*

■ Line 48: sentence starting with "Sparse and uneven…" could be a new paragraph.
**Response**: *The text starting with this sentence was divided into a new paragraph.*

■ Line 56: Consider changing "Recent researches on" to "Advances in"

**Response***: The words was replaced.*

Line 62: consider removing "methods such as" and changing "and other" to "with"
**Response***: The sentence was modified as "In addition, finding better predictors or combining SOM with other neural networks were also attempt to further decrease the pCO$_2$ predicting error".*

■ Line 68 " were considered as" to "are considered"
Line 74: "appeared" to "appears"
**Response***: The typos were corrected.*

■ Line 77: Consider writing sentence as "In addition, sampling information, such as latitude and longitude, has been used as a predictor." Also, consider referencing Gregor et al. 2019
**Response***: The sentence was modified as "In addition, sampling information, such as latitude and longitude and sampling time, has been used as a predictor." The reference was added.*

■ Lines 111-137: I would consider removing this text and moving Table S1 to the main text. Tables are much easier to read.
**Response***: Thank you for the suggestion. the Table S1 was moved to the main text.*

■ Line 145: Landschützer et al. (2020)
**Response***: The reference was corrected.*

■ Line 148: consider rephrasing "Provinces with connected pixels less than 10 and provinces with SOCAT observation less than 1000…" to "Provinces with less than 10 pixels and less than 1000 SOCAT observations…"
**Response***: The sentence was rephrased as the suggestion.*

■ Line 153: "The" needs to be lower case
Line 156: "200 m"
Line 167: comma after "summation".
Line 170: "a tan-sigmoid"
**Response***: The typos were corrected.*

■ Line 173: Maybe remove "pure" from "pure linear function", I am not sure what that means.
**Response***: The word was removed.*

■ Line 177: Please state how sensitive the model is to the choice of random number. That random number defines where the NN starts "searching" in

errors space. I understand the random number was fixed for reproducibility purposes. However, my understanding of best practice is to either run the model many times with different random numbers and take the average or "tune" the random number to find the one that gives best results. Please mention if this was explored or if the random number was simply chosen randomly.

**Response**: *Thank you for the suggestion. The fixed random number was currently chosen randomly. When using different random number streams, several predictors at the end of the output list of the stepwise FFNN algorithm differed, but the leading predictors were consistent, and the different predictors were also related. The fixed random number makes all networks using different predictors start training from the same point at the error space when comparing the performance of each predictor. In the future version, the method of taking the average will be used after we completely excluded the influence of initial state of FFNN on the results, and make sure the results would not change no matter which random number was used. Also, the algorithm may take much more time when taking the average to compare the perform of each predictor.*

- Line 224: "after that,"
- Line 226: "Then,"

**Response**: *The comma was added.*

- Line 231-235: Please consider rephrasing, this sentence is hard to understand.

**Response**: *The sentence was modified as "The part 1, including Loop 1, Selection step and Determine step 1 in Fig. 2, was repeated until no indicator was left in the Indicators pool or no decrease of $E_0$ can be found no matter which two indicators were added in the next two steps."*

- Line 236: "At this time, part 1 … finished and …"

**Response**: *The place of comma was corrected.*

- Line 259: In what increments where neurons increased by?

**Response**: *The neurons was set as [5,10,15,20,25,30,35,40,45,50,60,70,80,90,100,150,200,250,300]. The sentence was modified as "The number of neurons increased from 5 to 300 (the increment was five during 5-50 and ten during 50-100 and fifty during 100-300) and the corresponding MAE values of each size were recorded, and then the number of neurons with the lowest MAE was applied."*

- Line 290: It is also worth noting $pCO_2$ at BATS and HOT are estimated from TA and DIC (I am not sure if ESTOC is estimated or directly measured). I think it's important for the reader to know these estimates are derived and not directly measured like SOCAT observations.

**Response**: *Thank you for the suggestion. The sentence was modified as "The $pCO_2$ at HOT and BAT were estimated from TA and DIC, and $pCO_2$ at ESTOC were directly measured. These observations were not included in the SOCAT dataset."*

- Line 505: Wouldn't modifying the structure of the FFNN be considered making a "better network"? By better network do you mean more sophisticated architectures? I would also add that different learning algorithms could be considered.

**Response**: *Thank you for the suggestion. The sentence was modified as "A possible way to improve the performance of the stepwise FFNN algorithm is to modify the structure of FFNN or to use networks with more sophisticated architecture and to use different learning algorithms."*

- Table 4: Please state what the bold values indicate. Also a comma is needed before respectively: "…Landschützer et al., 2014 and Denvil-Sommer et al., 2019, respectively" This typo is repeated throughout the article

**Response**: *The lowest MAE and RMSE between different validation groups was shown in bold. The description was added and the typos was corrected.*

Figure 3: Make it clear these provinces are from a SOM. This is just a suggestion, but it would be great to get some of the information in Table 4 into this map. Maybe putting the leading predictor in parentheses in the colorbar? This might look too messy, it's just a suggestion.

**Response**: *The title of Figure 3 was modified as "The map of biogeochemical provinces based on SOM". It looks messy after adding the leading predictor, so Figure 3 was not changed.*

- Figure 6: Comma before "respectively"

Figure 7: Comma needed: "…Landschützer et al., 2014 and Denvil-Sommer et al., 2019, respectively

**Response**: *The typos were corrected.*

Figure 8: please put units on the colorbar.

**Response**: *The units were added.*

**Comments by reviewer Lucas Gloege**

I would like to thank the authors for the changes they made. I am appreciated that the authors took into account my previous comments. The manuscript is more clear and completed now.

Comments:

- Lines 38-44: New sentence is too long with repetitions, please modify it: "However, due to large uncertainty in estimates of surface ocean partial pressure of CO2 (pCO2), the long-term average global ocean sea-air CO2 flux during 2001-2015 estimated based on sea-air pCO2 difference differ from -1.55 to -1.74 PgC yr $^{-1}$, and the maximum difference between global sea-air CO2 flux in individual years reached nearly 0.6 PgC yr $^{-1}$ (Rödenbeck et al., 2014;Iida et al., 2015; Landschützer et al., 2014; Denvil-Sommer et al., 2019)". It can be something like: However, due to large uncertainty in estimates of surface ocean partial pressure of CO2 (pCO2), the long-term average global ocean sea-air CO2 flux during 2001-2015 differs from -1.55 to -1.74 PgC yr $^{-1}$ with the maximum difference in individual years nearly 0.6 PgC yr $^{-1}$ (Rödenbeck et al., 2014; Iida et al., 2015; Landschützer et al., 2014; Denvil-Sommer et al., 2019)."

**Response**: *The sentence was modified as "However, the global ocean sea-air $CO_2$ flux averaged between 2001-2015 varies from -1.55 to -1.74 PgC $yr^{-1}$ with the maximum difference in individual years nearly 0.6 PgC $yr^{-1}$, depending on the surface ocean partial pressure of $CO_2$ ($pCO_2$) product. These differences largely stem from differences in $pCO_2$ estimates across the products. (Rödenbeck et al., 2014; Iida et al., 2015; Landschützer et al., 2014; Denvil-Sommer et al., 2019)."*

- Line 123: Need a space between "Interim" and "(Dee et al…"

**Response**: *This part was changed to a table.*

- Line 128: You use here a word "parameters" to call "predictors" or "indicators". It appears further in the text as well (for example, in Conclusion, line 498). I think it can be misunderstood as parameters can be used to explain FFNN architecture. Please keep using words "predictors" or "indicators" throughout the text.

**Response**: *Thank you for the suggestion. The word "parameter" was all changed to "predictor".*

- Line 138: Please indicate which method you used for downscaling.

**Response**: *The products were downscaling by taking the average of all values in each 1° × 1° grid. This description was added in the text.*

- Lines 184-187: Please change this sentence, it is difficult to read: "The mean absolute error (MAE) difference that before and after adding or removing one indicator in the input of FFNN calculated using a K-fold cross validation method was used to estimate the performance of each indicator in the FFNN predicating."

**Response**: *The sentence was modified as "The mean absolute error (MAE), calculated using a K-fold cross validation method, was used to estimate the*

*performance of each predictor in the FFNN predicating."*

- Lines 318-320: Please combine these two sentences, it sounds like a repetition: "For example, month was considered as a recommended predictor in most provinces. Especially in the province P4 subpolar Atlantic and P5 north subtropical Atlantic, the parameter month was relatively more recommended."

**Response**: *The sentence was modified as "For example, the predictor month was considered recommended in most provinces, especially P4 subpolar Atlantic and P5 north subtropical Atlantic."*

- Line 322: Please add "the sine of latitude".

Line 404: Remove point between "Table" and "4".

**Response**: *Thank you for the suggestion. The typos were corrected.*